# Language-Instructed Vision Embeddings for Controllable and Generalizable Perception

**Chengzhi Mao, Xudong Lin, Wen-Sheng Chu**
Google
{czm, xudonglin, wschu}@google.com

## Abstract

Vision foundation models are typically trained as static feature extractors, placing the burden of task adaptation onto large downstream models. We propose an alternative paradigm: instead of solely feeding visual features into language models, we use language itself to dynamically guide the vision encoder. Our method, Language-Instructed Vision Embeddings (LIVE), leverages language as high-level guidance to produce task-centric embeddings at inference time, removing the need for task-specific retraining. This enables the encoder to focus on contextually relevant aspects of the input, yielding more controllable and generalizable representations. Empirically, LIVE reduces visual hallucinations (+34 points on MMVP), surpasses vision-language models with orders of magnitude more parameters on visual question answering, and generalizes to unseen instructions and tasks—offering a direct path toward adaptive, instruction-driven visual intelligence.

## 1 Introduction

A hallmark of human vision is its active, selective nature. Guided by internal goals or task demands, we focus on relevant parts of the visual world while ignoring distractions (Posner, 1980; Desimone et al., 1995). When searching for a specific object or understanding a particular interaction, humans implicitly "know" where and what to look for. In contrast, today's leading vision models, despite producing powerful general-purpose features (Oquab et al., 2023; Zhai et al., 2023; Yu et al., 2022; Radford et al., 2021), lack this dynamic, intention-driven adaptability. Their representations are typically *static*, pre-computed without reference to the specific query they are meant to serve.

This limitation is particularly acute in vision–language models. Existing approaches, such as visual prompting (Bahng et al., 2022; Shtedritski et al., 2023) or fine-tuning (Mao et al., 2022), offer limited adaptability; because they are optimized for specific target tasks, they fail to interpret zero-shot language instructions. Dominant LLM-centric architectures (Liu et al., 2023; Grattafiori et al., 2024; Alayrac et al., 2022; Team, 2024) delegate language integration to large downstream modules, incurring high computational cost while being unable to recover fine-grained details overlooked by the vision encoder, often resulting in hallucinations (Tong et al., 2024). Recent attempts to modulate vision encoders with paired captions (Lavoie et al., 2024; Xiao et al., 2025) are restricted by their reliance on descriptive text rather than true instructions, limiting their versatility and controllability. Thus, the central challenge remains: how to embed language-driven control into the vision encoder to yield adaptive, task-aware representations.

We address this challenge with LIVE (Language-Instructed Vision Embeddings), a simple and effective framework for creating **language-steered vision embeddings**. LIVE enables dynamic, fine-grained control of a vision encoder by training it to follow textual instructions. Concretely, we use a large language model (LLM) as the knowledge base to generate synthetic instruction–response pairs, which we combine with images into triplets. By teaching the vision encoder to steer embeddings based on textual prompts, the model can highlight task-relevant features while suppressing unrelated ones. For instance, in Figure 1, instructing the model to focus on 'fruit type' allows it to ignore typographical attacks, ensuring robust instruction-following directly within the representation space.

Once trained, LIVE yields standalone, language-steered embeddings that downstream tasks can use directly—no large LLMs or task-specific fine-tuning required. Trained on synthetic ImageNet-based data, LIVE generalizes strongly to real, unseen tasks: it reduces hallucinations by 34 points on

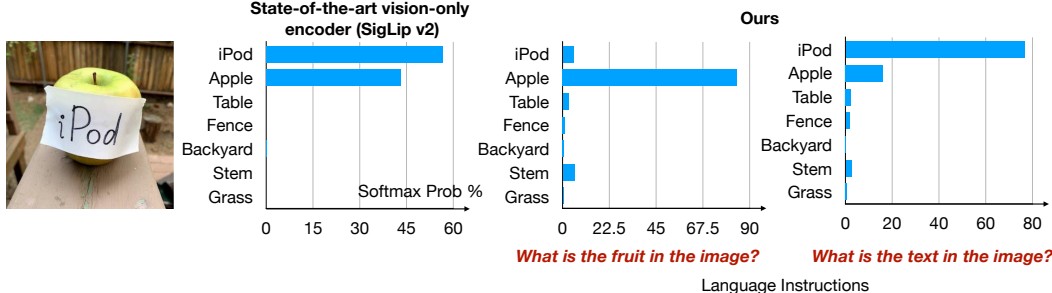

Figure 1: **LIVE (Language-Instructed Vision Embedding).** We show SoTA foundation model struggle to distinguish between textual content and objects within an image. LIVE enables user-guided attention to specified aspects (*e.g.*, "fruit" *v.s.* "text"), boosting control and prediction accuracy.

MMVP (Tong et al., 2024), and surpasses LLM-based counterparts on GQA (Hudson & Manning, 2019) by 7 points with less than 10% of their parameters. We also measure and narrow its gap to its LLM knowledge base by up to 49 points across instruction-following benchmarks. Attention and retrieval visualizations show precise, instruction-driven control emerging inside the encoder. With up to 10 times fewer parameters than LLM-heavy methods, our results suggest that embedding task instructions into the vision encoder—rather than scaling downstream modules—is an efficient path to generalizable, and controllable visual perception. More details and training data are available at our project page: `https://live-embedding.github.io/`.

## 2 RELATED WORK

**Vision Foundation Models.** Recent vision foundation models often use two-tower architectures and train contrastively with image-text pairs (Radford et al., 2021; Zhai et al., 2023; Tschannen et al., 2025; Zhai et al., 2022b). While some approaches jointly optimize contrastive and generative objectives (*e.g.*, CoCa (Yu et al., 2022), Mammut (Kuo et al., 2023)) or use encoder-based captioning (*e.g.*, Flamingo (Alayrac et al., 2022), Pali (Chen et al., 2022)), the vision embeddings are typically computed independently or language interaction occurs during late fusion transformers. Similarly, methods like Q-former (BLIP-2) (Li et al., 2022) use intermediate stages and powerful LLM decoders for image-to-text tasks, without directly instructing the frozen image encoder with language.

Alternative paradigms like masked image-text modeling (ViLT (Kim et al., 2021)) learn alignments but are not optimized for retrieval embeddings, therefore they require further finetuning on the target task and cannot perform prediction in a zero-shot manner. More recent architectures aim to unify approaches (X-Former (Swetha et al., 2024)) or leverage LLMs as decoders for richer outputs and supervision (Lin et al., 2021; Liu et al., 2023; Beyer et al., 2024; Team et al., 2025; Grattafiori et al., 2024; Wan et al., 2024; Tschannen et al., 2023; Lin et al., 2024). However, these methods still generally do not allow language to directly control the vision embeddings and cannot perform the target task via retrieval. Alternative vision only models, like Dino (Oquab et al., 2023; Caron et al., 2021) and JEPA (Assran et al., 2023), cannot handle language inputs. BRAVE (Kar et al., 2024) ensembles vision encoders for improved accuracy.

**Instructed Foundation Models.** The growing need for adaptive vision-language models inspired efforts in fine-tuning (Lin et al., 2023) and prompt engineering (Menon et al., 2022). However, these approaches typically optimize either the entire model or specific prompts, restricting them to single-task adaptations such as rationale explanation (Mao et al., 2023) or category classification (Mao et al., 2022). Methods like (Shtedritski et al., 2023; Zhong et al., 2022) allow querying visual encoders via explicit markers (*e.g.*, red circles or bounding boxes) but fail in scenarios involving overlapping or ambiguous visual concepts, as these markers only specify location without clarifying the targeted attribute (*e.g.*, color, texture). Prior work train top down vision encoder for embodied agent, yet this is not zero-shot (Eftekhar et al., 2023). Magiclens (Zhang et al., 2024) perform self-supervised image retrieval based on instructions, yet it does not provide retrieval in semantic, language space, and not ready for direct visual perception.

Multimodal retrieval methods like UniIR (Wei et al., 2024) perform retrieval via late-stage fusion of features, our work focuses on guiding the vision encoder itself, which could serve as an enhanced vision component to complement such models. (Kar et al., 2024) combines multiple vision encoders

Figure 2: **Instructive Vision Encoder Design.** Prior vision-language models such as CLIP (Radford et al., 2021) and SigLIP (Zhai et al., 2023) adopt two-tower architecture with separate vision and text encoders. We reuse the text tower to encode the query, apply a projection layer, and inject it into the vision transformer alongside with the input image. Note that although the question and the answer are passed through the same text encoder, they are processed independently with no feature interaction between them. The text encoder remains frozen during training, while only the vision encoder is updated (highlighted in yellow). Learnable embeddings are shown in pink.

to obtain better vision representations for language models. Other approaches control vision indirectly through post-hoc modification (Chen et al., 2024a) or in specialized domains like document retrieval (Zhou et al., 2024; Chen et al., 2024b). A recent trend is fine-tuning vision LLMs for retrieval (Wei et al., 2024; Jiang et al., 2024; Liu et al., 2025). While powerful, these models inherit the substantial computational footprint of their underlying LLMs. The most related works that also modulate the vision encoder directly typically use image captions as the conditioning signal (Lavoie et al., 2024; Xiao et al., 2025). This strategy risks learning undesirable shortcuts, as the model can minimize loss by simply matching text features rather than learning true visual grounding. Our method, LIVE, explicitly decouples the guidance from the target by using task instructions that differ significantly from the target description. This design forces the model to learn a more sophisticated mechanism for instruction-based control, enabling precise manipulation of vision embeddings without the inference overhead of large LLMs or the risk of learning trivial solutions.

## 3 METHODOLOGY: LEARNING LANGUAGE-INSTRUCTED VISION EMBEDDINGS

Conventional vision-language models treat the vision encoder as a static feature extractor for a downstream LLM. We reverse this paradigm by distilling knowledge from an LLM back into the vision encoder. Using synthetic image-query-answer triplets, we train the vision encoder with contrastive learning to produce language-instructed embeddings that align with the answer's semantics. The result is a powerful, standalone vision encoder capable of zero-shot perception, removing the need for a computationally expensive LLM at inference.

### 3.1 LANGUAGE-INSTRUCTED VISUAL EMBEDDINGS

Standard vision-language models like CLIP (Radford et al., 2021) and SigLIP (Zhai et al., 2023) use a two-tower architecture with vision $E(\cdot)$ and text encoders $T(\cdot)$ (see Figure 2 for an illustration). The vision encoder $E$ produces a general-purpose embedding $\mathbf{z} = E(\mathbf{x})$ designed to capture all relevant information from the input image $\mathbf{x}$. However, to support diverse downstream tasks via text queries, these embeddings must be both versatile and precise. Learning such universal representations is challenging due to the inherent capacity limits of the vision encoder.

Visual prompting (Bahng et al., 2022; Jia et al., 2022) seeks to adapt visual representations. However, visual perception is often ambiguous and context-dependent. Simple location prompts (*e.g.*, boxes, circles (Shtedritski et al., 2023)) offer limited control and cannot specify which aspects to focus on. For instance, a car region might require attention on color ("What color?"), make ("What make?"), or condition ("Is it clean?"). Existing methods struggle to handle this kind of semantic ambiguity.

To address this, we propose a language-conditioned vision encoder, denoted $E_{live}$. Instead of a fixed embedding, $E_{live}$ dynamically processes the image $\mathbf{x}$ based on the embedding of a textual instruction $\mathbf{q}$. We reuse the pretrained text encoder $T(\cdot)$. Our instructive visual embedding is computed as:

$$\mathbf{z}^{(I)} = E_{live}(\mathbf{x}, T(\mathbf{q})). \tag{1}$$

| | Question | Answer |
|---|---|---|
| | *What is the primary activity taking place in the image?* | A person is driving a snowmobile with a child passenger. |
| | *What type of vehicle is depicted in the image?* | A vintage snowmobile. |
| | *What is the color of the snowmobile?* | The snowmobile is predominantly blue and white. |
| | *What is the color of the child's jacket?* | Orange |
| | *What is the adult wearing on their head?* | A helmet. |
| | *What is visible in the background of the image?* | Houses, trees, a road, and a bright sun |
| | *What is the license plate number on the snowmobile?* | 542958 |
| | *What is the color of the mug in the center of the image?* | The mug in the center is primarily yellow and brown. |
| | *What character is featured on the yellow mug?* | The yellow mug features Donald Duck. |
| | *What object is behind the mugs?* | A coffee maker is behind the mugs. |
| | *What is the relative position of the red mug to the yellow?* | The red mug is to the left of the yellow mug. |
| | *Is there coffee in any of the mugs?* | Yes, there appears to be coffee in the yellow mug. |
| | *What kind of surface are the mugs sitting on?* | The mugs are sitting on a white shelf or counter. |
| | *If the foreground is ignored, what is the main object in the image?* | If the foreground is ignored, the main object in the image is the coffee maker |

Figure 3: **Triplet Training Data from LLM.** We apply Gemini-2.0-Flash (Comanici et al., 2025) to automatically generate diversified, open-world triplet data containing image, query, and answer. This method moves beyond generic questions from existing image-text datasets, enabling more nuanced and sophisticated exploration of image-specific content.

This formulation enables the vision encoder to focus on the aspects of the image most relevant to the language instruction, producing a targeted, task-specific representation. Model implementation details are provided in Section 4.1.

## 3.2 Training Objective

We train the instruction-conditioned vision encoder $E_{live}$ by matching its output $\mathbf{z}^{(I)}$ with the text embedding of the corresponding correct answer $\mathbf{a}$. Specifically, we want our instructed vision embedding $\mathbf{z}_i^{(I)} = E_{live}(\mathbf{x}_i, T(\mathbf{q}_i))$ to be close to the answer text embedding $\mathbf{z}_j^{(T)} = T(\mathbf{a}_j)$ if and only if $(\mathbf{x}_i, \mathbf{q}_i)$ corresponds to answer $\mathbf{a}_j$.

Following (Zhai et al., 2023), we employ a sigmoid-based alignment loss, which yields better performance and stability than standard contrastive losses (Radford et al., 2021). Given a batch of image-instruction pairs $(\mathbf{x}_i, \mathbf{q}_i)$ and their corresponding answers $\mathbf{a}_j$, the loss is defined as:

$$\mathcal{L} = -\mathbb{E}_{i,j}\left[\log \frac{1}{1 + \exp\left(-y_{ij}(t(\mathbf{z}_i^{(I)} \cdot \mathbf{z}_j^{(T)}) + b)\right)}\right]. \tag{2}$$

$y_{ij} \in \{-1, 1\}$ encodes the match of the image-query-answer triplet (1 for match, $-1$ for mismatch). The parameters $t$ (temperature) and $b$ (bias) are learnable parameters for calibration. We optimize the visual encoder $E_{live}$ by minimizing this loss with gradient descent.

## 3.3 Knowledge Transfer from LLM

Despite the abundant image text paired data (Schuhmann et al., 2022; Byeon et al., 2022), a significant challenge in training the instruction-guided encoder $E_{live}$ is the scarcity of large-scale datasets with image-instruction-answer triplets $(\mathbf{x}, \mathbf{q}, \mathbf{a})$. Existing visual question answering (VQA) datasets (*e.g.*, CC3M-VQA (Changpinyo et al., 2022)) often rely on template-based or rule-generated questions, which may not capture the breadth and complexity of real-world queries needed to probe deeper understanding. Our experiments in Figure 6 shows existing datasets are insufficient for training language-instructed visual embeddings with high accuracy.

To overcome this data bottleneck, we leverage the extensive world knowledge and reasoning capabilities in LLMs. We treat an LLM as an implicit knowledge source that can identify salient aspects of an image and generate relevant questions about them. Specifically, we query powerful LLMs that accept visual inputs to generate question-answer pairs $(\mathbf{q}, \mathbf{a})$ conditioned on the given image. This effectively transfers the LLM's rich understanding from billions of parameters into training data for our vision encoder. Crucially, this computationally intensive LLM inference is performed offline

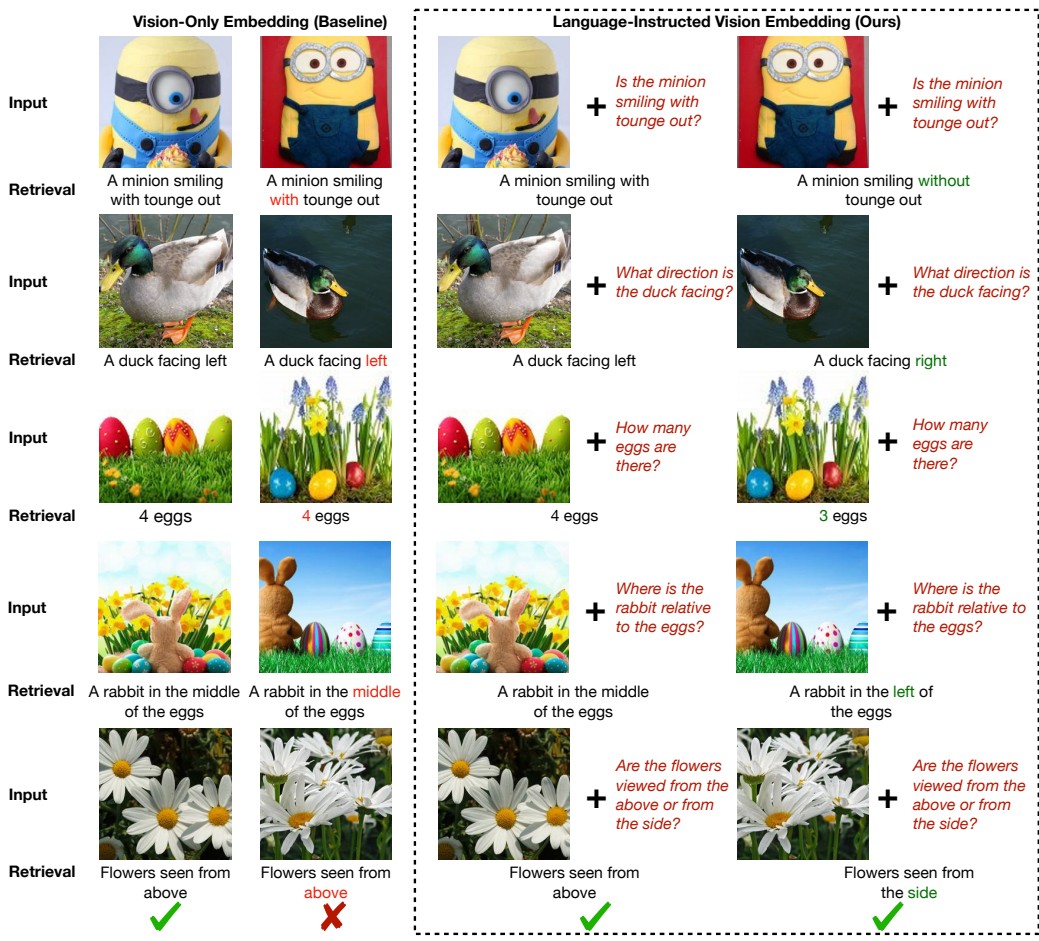

Figure 4: **LIVE Reduces Visual Hallucinations (MMVP Benchmark Tong et al. (2024)).** State-of-the-art vision-only embeddings Zhai et al. (2023) (left column) encode the entire scene without query-specific guidance, making them prone to hallucination when fine-grained precise details are required. By modulating visual computation with the input text query (right column), our method selectively focuses on relevant information, thereby reducing hallucinations and improving accuracy.

during dataset creation. At inference, only the efficient instruction-guided vision encoder $E_{live}$ is required, maintaining real-time computational efficiency for perception tasks.

For each image, we prompt LLM to generate multiple diverse question-answer pairs using the following prompt structure: *Provide a numbered list of interesting visual questions about the image, followed by the corresponding answers.* Figure 3 shows examples of our generated queries and answers on ImageNet, introducing diverse visual attributes and semantic concepts that were previously unavailable. This rich, detailed data enables vision models to learn beyond prevalent image-captioning patterns, fostering more effective and fine-grained visual comprehension. Importantly, humans often perform such queried visual tasks using System-1 (intuitive) reasoning. Our LIVE encoder is designed to operate with similar efficiency to avoid the computational overhead of larger LLMs, especially visual perception applications. Moreover, when downstream tasks are known before deployment, our text embeddings can be pre-computed and cached to further save computation costs.

## 4 EXPERIMENT

This section details our experimental setup, benchmarks, baselines, results, and analysis designed to evaluate the zero-shot language controllability enabled by our LIVE approach.

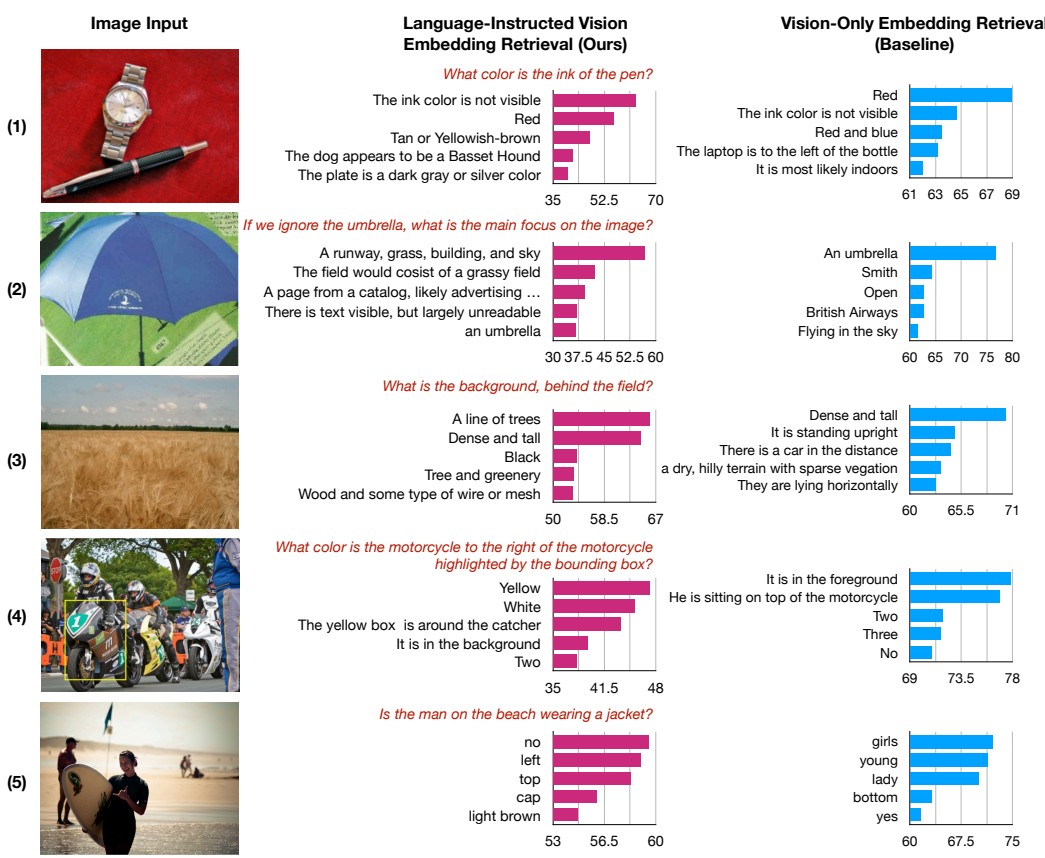

Figure 5: **LIVE's Retrieval based on Language Instructions.** Examples 1-5 show examples from ImageNet, Caltech, SUN, RefCOCO, and GQA, respectively. The instructions provided at inference time are unseen during training and highlighted in *red*. For both our method and the vision-only baseline SigLIP (Zhai et al., 2023), we show the top-5 retrieved text responses with bars indicating the predicted sigmoid probabilities. Our method demonstrates superior retrieval accuracy, as it correctly (1) identifies non-visible elements, (2) follows instructions to ignore specified content, (3) attends to the requested factors, (4) performs basic spatial reasoning, (5) captures relationships between objects.

## 4.1 EXPERIMENTAL SETUP

**Training Data.** We train LIVE using ImageNet training images, and apply the data generation process described in Section 3.3 using ImageNet dataset and Gemini 2.0 Flash[1] (Gemini, 2024; Comanici et al., 2025). This yields ∼16.4 million images-query-answer triplets, as multiple informative queries are generated per image. We also explore and compare against publicly available triplet data from PaliGemma (Beyer et al., 2024), specifically leveraging CC3M-VQA (Changpinyo et al., 2022). For datasets without explicit instructions, such as WebLI (Wang et al., 2025) and Open Images (Piergiovanni et al., 2022), we introduce a universal instruction, "caption the image".

**Evaluation Benchmarks.** We target tasks that require explicit language instructions to define the objective, in contrast to conventional benchmarks that rely on static vision encoders for universal zero-shot classification, and thus cannot evaluate instruction-following or capabilities beyond fixed label taxonomies. We deduplicate both images and text instructions to ensure all our evaluations data are novel. For all datasets, we report Top-1 retrieval accuracy. MMVP (Tong et al., 2024) is a recent benchmark designed to evaluate hallucinations in VLMs (see example in Figure 4). It pairs visually similiar images so that models are required to attend to the nuances to answer correctly. GQA (Hudson & Manning, 2019) is a challenging visual question answering benchmark that extends beyond attribute answering and requires reasoning over scene graphs to answer relational queries.

---

[1]ImageNet images https://www.image-net.org/ and Gemini 2.0 Flash API are publicly available. The generated query-answer pairs are provided on our project webpage.

| | Image Size | Params (M) | 🧭 | 🔍 | 🔄 | 🔢 | 📍 | 🎨 | ⚙️ | A | 📷 | Average |
|---|---|---|---|---|---|---|---|---|---|---|---|---|
| OpenAI ViT-L-14 (Radford et al., 2021) | $224^2$ | 427.6 | 13.3 | 13.3 | 20.0 | 20.0 | 13.3 | 53.3 | 20.0 | 6.7 | 13.3 | 19.3 |
| OpenAI ViT-L-14 (Radford et al., 2021) | $336^2$ | 427.9 | 0.0 | 20.0 | 40.0 | 20.0 | 6.7 | 20.0 | 33.3 | 6.7 | 33.3 | 20.0 |
| DFN ViT-H-14 (Fang et al., 2023) | $224^2$ | 986.1 | 20.0 | 26.7 | 73.3 | 26.7 | 26.7 | 66.7 | 46.7 | 13.3 | 53.3 | 39.3 |
| DFN ViT-H-14 (Fang et al., 2023) | $378^2$ | 986.0 | 13.3 | 20.0 | 53.3 | 33.3 | 26.7 | 66.7 | 40.0 | 20.0 | 40.0 | 34.8 |
| MetaCLIP ViT-L-14 (Xu et al., 2023) | $224^2$ | 427.6 | 13.3 | 6.7 | 66.7 | 6.7 | 33.3 | 46.7 | 20.0 | 6.7 | 13.3 | 23.7 |
| MetaCLIP ViT-H-14 (Xu et al., 2023) | $224^2$ | 986.1 | 6.7 | 13.3 | 60.0 | 13.3 | 6.7 | 53.3 | 26.7 | 13.3 | 33.3 | 25.2 |
| EVA01 ViT-g-14 (Sun et al., 2023) | $224^2$ | 1136.4 | 6.7 | 26.7 | 40.0 | 6.7 | 13.3 | 66.7 | 13.3 | 13.3 | 20.0 | 23.0 |
| EVA02 ViT-bigE-14+ (Sun et al., 2023) | $224^2$ | 5044.9 | 13.3 | 20.0 | 66.7 | 26.7 | 26.7 | 66.7 | 26.7 | 20.0 | 33.3 | 33.3 |
| SigLIP ViT-SO-14 (Zhai et al., 2023) | $224^2$ | 877.4 | 26.7 | 20.0 | 53.3 | 40.0 | 20.0 | 66.7 | 40.0 | 20.0 | 53.3 | 37.8 |
| SigLIP ViT-SO-14 (Zhai et al., 2023) | $384^2$ | 878.0 | 20.0 | 26.7 | 60.0 | 33.3 | 13.3 | 66.7 | 33.3 | 26.7 | 53.3 | 37.0 |
| InstructBLIP (Dai et al., 2023) | $336^2$ | ~14200.0 | - | - | - | - | - | - | - | - | - | 16.7 |
| LLaVa (Liu et al., 2023) | $336^2$ | ~13000.0 | - | - | - | - | - | - | - | - | - | 31.3 |
| BRAVE Kar et al. (2024) | $336^2$ | ~10300.0 | - | - | - | - | - | - | - | - | - | 42.0 |
| SigLIP ViT-SO-14 (Ours) | $384^2$ | 891.0 | **80.0** | **76.7** | **73.3** | **80.0** | **83.3** | **86.7** | **66.7** | **66.7** | **73.3** | **76.3** |

Table 1: **Zero-Shot Accuracy on MMVP-VLM benchmark (Tong et al., 2024).** We use **bold** to highlight the highest accuracy. Baseline methods are vision-only models, with their results quoted from MMVP. Following MMVP, we denote visual patterns as follows. 🧭: Orientation and Direction, 🔍Q: Presence of Specific Features, 🔄: State and Condition, 🔢: Quantity and Count, 📍: Positional and Relational Context, 🎨: Color and Appearance, ⚙️: Structural and Physical Characteristics, **A**: Texts, 📷: Viewpoint and Perspective. All CLIP-based methods using vision-only embeddings struggle on this benchmarks. By incorporating instruction-guided modulation, our method achieves a **34-point** zero-shot accuracy improvement over prior SOTA methods, highlighting the role of instructions in directing the vision encoder towards relevant signals and reducing hallucinations.

(a) Performance of our model variants on GQA.

| ViT Model | SigLIP | Fusion | Menon *et al.* | **Ours** |
|---|---|---|---|---|
| SigLIP ViT-T-14 | 9.8 | 20.0 | 10.4 | **60.6** |
| SigLIP ViT-B-16 | 12.0 | 12.8 | 13.0 | **71.2** |
| SigLIP 2 ViT-B-16 | 14.4 | 20.8 | 19.8 | **67.6** |
| SigLIP 2 ViT-SO-14 | 16.4 | 19.6 | 17.8 | **68.2** |

(b) Comparison with SoTA methods.

| Model | Accuracy (%) |
|---|---|
| BLIP-2 (Li et al., 2022) | 44.7 |
| InstructBLIP (Dai et al., 2023) | 49.5 |
| BRAVE (Kar et al., 2024) | 52.7 |
| LLava (Liu et al., 2023) | 63.3 |
| **Ours (ViT-B-16)** | **71.2** |

Table 2: **Zero-Shot Retrieval Accuracy on GQA tasks.** We report top-1 accuracy (%).

To quantify the gap between LIVE and its LLM knowledge source, we use Gemini 2.0 Flash to annotate instruction answers on test data from Caltech101 (Griffin et al., 2007), SUN397 (Xiao et al., 2010), RefCOCO (Kazemzadeh et al., 2014), and ImageNet (Deng et al., 2009). Under this setup, Gemini 2.0 Flash achieved 100% accuracy by construction. We filter the test data to ensured no instruction overlap between training and test sets, and denote the repurposed datasets as †. Those † datasets are intended to measure the fidelity of knowledge transfer from the "teacher" (Gemini) to the "student" (LIVE), rather than benchmark performance on the original target tasks, since those instructions are Gemini-generated.

**Baselines.** We consider a family of static vision-only embedding baselines and their variants. *SigLIP* (Zhai et al., 2023): We use publicly available SigLIP models up to size SO400M, representing the SoTA static, instruction-agnostic visual embeddings. *Fusion*: We directly add the image and text query embeddings, and use the combined representation to retrieve the text answer. *Menon et al.* (Menon & Vondrick, 2022): Following Menon *et al.*, who improved retrieval by augmenting text answers with language descriptions, we append the language instructions to the answer so that the text tower is explicitly informed of the query. We further compare against LLM-based approach *LLaVa (Liu et al., 2023)*, late fusion methods *InstructBLIP (Li et al., 2022)*, and an ensemble-based state-of-the-art method *BRAVE (Kar et al., 2024)* that combines five generic vision encoders EVA-CLIP-g (Sun et al., 2023), CLIP-L/14 (Radford et al., 2021), SILC-G/16 (Naeem et al., 2024), ViT-e (Chen et al., 2022), and DINOv2-L/14 (Oquab et al., 2023) followed by further fine-tuning.

**Implementation Details.** We initialize LIVE's vision encoder from a pretrained SigLIP and SigLIP-v2 (Zhai et al., 2023; Tschannen et al., 2025), which outperform CLIP (Radford et al., 2021). All models are based on the transformer architecture (Vaswani et al., 2017). We use SigLIP text encoder to precompute fixed instruction embeddings for both training and evaluation. These embeddings are

| ViT Model | ImageNet† | | | | Caltech 101† | | | |
|---|---|---|---|---|---|---|---|---|
| | SigLIP | Fusion | Menon et al. | Ours | SigLIP | Fusion | Menon et al. | Ours |
| SigLIP ViT-T-14 | 25.10 | 32.46 | 33.42 | **73.28** | 10.53 | 11.38 | 22.05 | **37.08** |
| SigLIP ViT-B-16 | 30.84 | 33.23 | 42.50 | **86.93** | 12.08 | 12.64 | 24.72 | **55.75** |
| SigLIP 2 ViT-B-16 | 37.73 | 40.69 | 60.52 | **86.79** | 14.89 | 15.31 | 29.92 | **51.97** |
| SigLIP2 ViT-SO-14 | 38.03 | 40.40 | 60.86 | **87.06** | 14.61 | 15.87 | 33.00 | **55.05** |
| ViT Model | SUN† | | | | RefCOCO† | | | |
| | SigLIP | Fusion | Menon et al. | Ours | SigLIP | Fusion | Menon et al. | Ours |
| SigLIP ViT-T-14 | 6.99 | 8.87 | 10.90 | **33.16** | 8.52 | 11.74 | 11.01 | **42.73** |
| SigLIP ViT-B-16 | 9.26 | 9.75 | 16.94 | **49.83** | 9.84 | 10.87 | 12.78 | **59.32** |
| SigLIP 2 ViT-B-16 | 12.44 | 12.96 | 24.67 | **49.76** | 12.04 | 13.51 | 17.47 | **55.95** |
| SigLIP 2 ViT-SO-14 | 13.00 | 14.06 | 25.79 | **52.94** | 9.40 | 10.28 | 14.98 | **54.33** |

Table 3: **Closing the gap to the Gemini knowledge source in zero-shot instruction following.** We report Top-1 retrieval accuracy on benchmarks †, where Gemini's annotations act as a 100% accurate oracle. Our model is evaluated in a strict zero-shot setting, without any fine-tuning on the downstream Caltech 101, SUN, or RefCOCO datasets, unlike prior work (Beyer et al., 2024; Kim et al., 2021). All evaluation data are deduplicated from our synthetic training set. Our approach substantially narrows the gap to the oracle, outperforming baselines by up to 49 points.

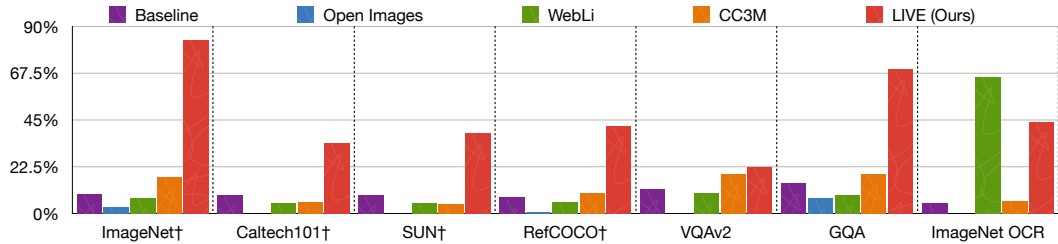

Figure 6: **Impact of triplet training data on LIVE's accuracy.** We train SigLIP v2 ViT-B-16 with four triplet datasets, Open Images, WebLI, CC3M, and ours. Ours achieves broader improvements across benchmarks. While Open Images showed no gain, WebLI increases OCR, and CC3M offered slight improvements on some tasks, our approach highlights the benefit of using LLMs to overcome traditional data limitations for training transferable vision encoders.

projected through a single linear layer and injected into the vision transformer, which adds ∼13M parameters for the ViT-So model. During training, the vision encoder, including the text projection layer, is updated, while the original text tower remains frozen. We adopt the same optimizer setting as SigLIP (Zhai et al., 2022a) with a learning rate of 0.001, batch size of 8192, and 122k training steps, using 256 TPUv3 cores. As in SigLIP, we apply only resize augmentation during training.

## 4.2 RESULTS

We first evaluate LIVE on the MMVP-VLM benchmark (Tong et al., 2024). As shown in Table 1, our model achieves a 34-point accuracy gain over prior methods, including LLM-based and ensemble approaches that are up to 10 times larger. Qualitative examples in Figure 4 illustrate how language instructions guide the vision encoder towards task-relevant cues, thereby reducing hallucinations (*e.g.*, incorrectly perceiving a minion's tongue). On the GQA benchmark, which requires additional reasoning, LIVE outperforms both LLMs and the strongest generic vision models by 7 points, while using 10 times fewer parameters.

We further measure LIVE's accuracy gap to its LLM knowledge source, Gemini 2.0 Flash. As shown in Table 3, although a 23-41 point gap to the Gemini oracle remains, LIVE consistently attains Top-1 retrieval accuracy over established vision-only embeddings on these targeted tasks, despite considerable domain shifts in both images and query types.

Figure 5 shows visualizations of images, queries, and top-5 retrieved instances, all deduplicated from the training set. Our LIVE model exhibits emergent capabilities beyond its training data. For instance, image (4) shows the model correctly interprets bounding boxes to identify the color of

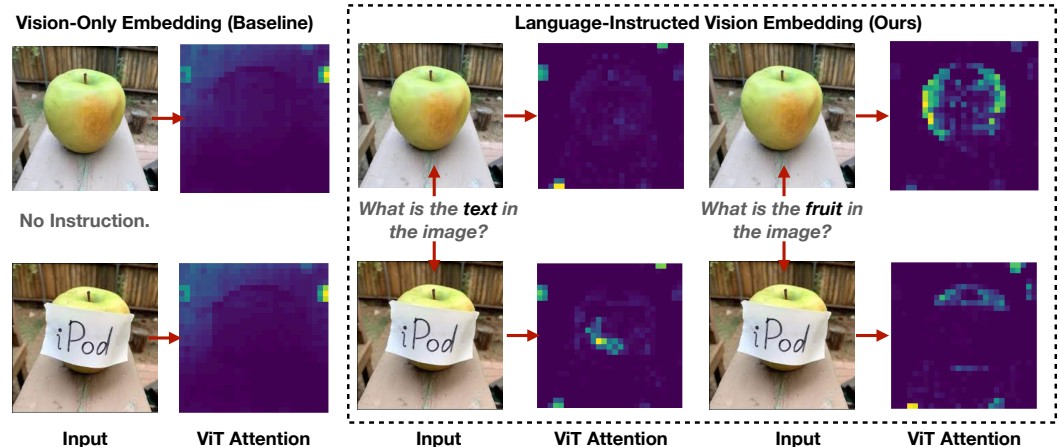

Figure 7: **Zero-Shot Language Instructions Steer Visual Attention.** Unlike baseline encoders producing global attention (SigLIP, left), our LIVE uses instructions to focus dynamically. Prompting for "text" highlights the "iPod" label; prompting for "fruit" highlights only the apple, ignoring the label. This demonstrates emergent, instruction-driven control over visual encoding.

| Training Groups
Testing Groups | SVD
F | FVD
S | FSD
V | FSV
D | FSVD
FSVD |
|---|---|---|---|---|---|
| SigLIP 2 ViT-B/16 | 74.05 | 82.48 | 83.40 | 83.28 | 86.93 |

Table 4: **Leave-One-Group-Out Generalization.** To test generalization to novel instruction types, we partition our data into four categories: Fundamental Properties (F), Spatial-Textual (S), Viewpoint (V), and Dynamic Reasoning (D). We train the model while holding out each category in turn, demonstrating LIVE's ability to generalize to semantically distinct, unseen instructions.

a specified motorcycle, despite never seeing bounding box annotations during training. Similarly, image (1) highlights the model's ability to reason about nuanced visual details, such as recognizing that an ink color is not visible, rather than defaulting to the image's dominant red color (as seen with baseline vision-only embedding). These examples illustrate fine-grained understanding and contextual reasoning previously unattainable with static vision-only embeddings.

## 4.3 ANALYSIS

**Impact of Training Data.** We benchmark our model by training it individually on established vision-language datasets, I.e., Open Images (Piergiovanni et al., 2022), WebLI (Wang et al., 2025), CC3M-VQA (Changpinyo et al., 2022), and on our newly constructed Imagenet triplet dataset. For datasets lacking explicit textual queries (unlike CC3M-VQA, which provides rule-extracted query-answer pairs), we employed generic queries (*e.g.*, "caption the image") to ensure a comparable training setup. As shown in Figure 6, models trained with our Imagenet triplet dataset significantly outperform those trained on existing image-language datasets across diverse benchmarks. These results indicate that the lack of large-scale, diverse, and targeted image-query-answer data has been a major bottleneck for advancing instruct-aware vision embeddings. While prior work typically freezes vision model and improves the LLM, we reverse this paradigm and show that LLMs can be instead leveraged to more effectively train vision models.

**Impact of Vision Encoder Size.** As detailed in Table 3, we scale the vision encoder from ViT-T (5.4M parameters) to ViT-B (86.6M) and SO400M (891M). Although performance improves with model size, the compact ViT-T model still attains competitive accuracy, indicating its potential for deployment on resource-constrained edge devices.

**Attention Map Visualization.** Figure 7 visualizes how language instructions modulate visual attention. We plot heatmaps of the attention from our language input token to the visual tokens. We contrast a baseline vision-only transformer (SigLIP ViT-SO-14, left) with ours (right). Given the same

| ViT-B | GQA | MMVP |
|---|---|---|
| Layer 1 | 67.4 | 69.5 |
| Layer 4 | 67.8 | 69.4 |
| Layer 8 | 68.2 | 68.7 |

Table 5: **Impact of Language Injection Depth (ViT-B):** Late injection favors GQA, implying relation-heavy tasks require higher-level semantics, while early injection better reduces visual hallucinations in MMVP. Ultimately, no single insertion point is universally optimal across all benchmarks.

| Method | Text Query (Vision Input) | Text Target | GQA | MMVP |
|---|---|---|---|---|
| Ablation 1 | Neutral ("Caption the image.") | Rich Answer | 13.1 | 65.1 |
| Ablation 2 | Specific ("What is the category of the image?") | Class Name | 2.7 | 54.7 |
| LIVE (Ours) | Specific Query | Rich Answer | **67.4** | **69.5** |

Table 6: **Impact of Text in Guiding Vision Representation.** We change the text query and target to isolate the effects of instruction specificity and supervision granularity.

input image (*e.g.*, an apple labeled as "iPod"), the baseline's attention remains instruction-agnostic, as it does not condition on instructions. In contrast, LIVE dynamically adjusts its focus: when instructed to find "the text", attention focuses on the "iPod" label; when asked to identify "the fruit", attention localizes on the apple. This demonstrates that LIVE learns to steer its visual processing according to the language query, enabling focused computation on instruction-relevant regions.

**Generalization to Out-of-Distribution (OOD) Instructions Groups.** To conduct a stricter generalization test, we train and evaluate our model on semantically disjoint instruction groups. This introduces a larger distributional shift than the deduplication strategy used in prior experiments. As shown in Table 4, the model maintains strong performance even under this challenging OOD setting.

**Impact of Language Injection Depth.** We evaluate the effect of injecting language tokens at different depths (Layer 1, 4, and 8) of the ViT-B encoder. As shown in Table 5, early injection (Layer 1) yields the highest performance on MMVP (69.5), where preserving fine-grained visual details is critical for detecting hallucinations. In contrast, late injection (Layer 8) performs better on GQA (68.2), suggesting that relation-centric tasks benefit from higher-level semantic abstraction. These results indicate language tokens actively modulate the visual features at different processing stages.

**Impact of Text in Guiding Vision Representation.** To validate that instruction conditioning—not simply data scale—-drives performance, we ablate the encoder input types (Table 6). Replacing LIVE's specific queries with neutral prompts (*e.g.*, "Caption the image") collapses GQA accuracy from 67.4 to 13.1, showing that vision encoders require targeted, query-driven guidance. Moreover, replacing rich, descriptive answers with standard class labels drops GQA performance to near-random levels (2.7). Together, these results confirm that LIVE's gains stem from explicit language-to-vision instruction conditioning and descriptive supervision, rather than from the backbone architecture or dataset scale alone.

## 5 CONCLUSION

We introduce a new paradigm for vision representation: instructing the vision encoder with knowledge from language models. Unlike conventional approaches that freeze a generic vision encoder, we show injecting task-specific guidance directly into the visual system provides substantial benefits. Our method produces an efficient, lightweight encoder that improves perceptual precision and mitigates hallucinations without costly retraining. Our findings suggest that advancing vision models on targeted tasks relies not only on scaling, but also on making them instruction-aware.

## ACKNOWLEDGEMENTS

We would like to thank Guangxing Han for their invaluable insights and discussions, and Longqi Cai for crucial infrastructure support. We are also deeply grateful to Yaojie Liu for executing all the key experiments during the rebuttal phase, and Ahmed Abdelkader for providing feedback for the paper.

ETHICS STATEMENT

Our work introduces instruction-aware vision encoders that accept natural-language task specifications. While this can reduce hallucination and improve task precision, it also raises ethical considerations: Instruction following could be repurposed for harmful objectives (*e.g.*, surveillance, targeted profiling). In our training, we do not include any harmful objectives, therefore the risk shall be minimized in our model perspective.

REPRODUCIBILITY STATEMENT

To ensure reproducibility, we have provided comprehensive implementation details, network architectures, and hyperparameter configurations throughout the paper and the Appendix. Because the original training codebase is deeply integrated with proprietary internal infrastructure, it cannot be directly open-sourced. Furthermore, the release of the training dataset is currently undergoing internal institutional review. Pending final open-source approval, the dataset will be made publicly available.

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

# A   APPENDIX

## A.1   LIMITATIONS

Our approach enhances the controllability of visual representations using language instructions. However, its practical application and further development are subject to certain limitations, which also open avenues for future research.

**Optimizing Query Design for Downstream Tasks:** A primary challenge lies in the formulation of effective textual queries to maximize performance on specific downstream applications. The process of identifying the optimal phrasing, level of detail, and linguistic structure for queries that elicit the desired visual representation changes remains an empirical endeavor. It may require significant tuning for each new task or dataset. This is compounded by the inherent ambiguity and richness of natural language, where subtle variations in a query can lead to different steering outcomes, not all of which may be beneficial for the target application's accuracy. We conducted initial experiments in Figure 12, yet a principled way to discovery effective prompt is still missing.

**Handling of Complex and Compositional Queries.** The reliance on a pretrained text encoder constrains the complexity of queries our method can effectively interpret. Current pretrained text encoders, while powerful, often struggle with deeply compositional or abstract textual prompts. Their encoding of nuanced relationships between multiple concepts, or negation, might not be robust. Our method, therefore, performs best with relatively simple, direct queries.

**Potential for Undesired Steering Outcomes.** Depending on how the users provide the instructions, the model has a risk of generating biased, unsafe, or undesirable content.

## A.2   FUTURE WORK

**Principled Query Optimization and Discovery:** Developing systematic methods or even learnable components to automatically discover or refine queries for optimal downstream performance would significantly enhance usability. This could involve techniques from prompt engineering, reinforcement learning, or semantic search to bridge the gap between user intent and effective query formulation.

**Enhancing Complex Query Understanding:** Future work should focus on strategies to decompose complex textual queries into simpler, manageable sub-queries that our current framework can process. Alternatively, exploring new architectures or fine-tuning regimes for the text encoder to better handle compositional semantics and logical operations directly within the query embedding space would be a valuable pursuit. This could involve incorporating structured knowledge or symbolic reasoning alongside neural representations.

**Visual Grounding with Instructions:** Our method can mitigate visual hallucinations, which can be used as a component in RAG systems, to help verify, ground the reasoning and prediction of LLM.

**Language-Instructed Vision Generation:** Our method is a language-instructed vision encoder, which can be used as the backbone that encode semantic information in generative models, such as Diffusion. For example, by using language-instructed vision embeddings, one can train image editing models based on the instructions.

## A.3   BROADER IMPACT

Our research on language-steered vision embeddings has the potential for considerable positive societal impact, primarily by offering novel approaches to creating more equitable and robust AI systems. By enabling vision embeddings to be guided by language instructions, we introduce a mechanism for actively mitigating biases present in training datasets. This zero-shot bias mitigation capability is a significant step towards fairer AI representations, as it allows for targeted adjustments without the need for extensive retraining or dataset curation, making the development of equitable models more accessible.

Furthermore, our method enhances the robustness of vision embeddings. By training models on detailed, instructed triplets, they learn to capture nuanced, fine-grained signals from an image, moving beyond a single, holistic embedding. This improved granularity can lead to models that are more adaptable and less susceptible to being misled by irrelevant or superficial features. An important

application of this enhanced instructional control is the ability to direct the model to defend against typographical attacks. This contributes to making vision models safer and more resilient to adversarial manipulations aimed at "jailbreaking" or deceiving them.

However, we also recognize potential negative societal impacts. The same linguistic steerability that allows for bias mitigation and robustness enhancement could, if misused, be employed to intentionally introduce or amplify biases. A malicious actor could craft instructions to make the vision embeddings unfairly prejudiced against certain groups or characteristics. Currently, our work does not include a mechanism to automatically discriminate between benign and malicious instructions, nor a system to refuse potentially harmful guidance. This creates a risk of misuse, where the technology could be exploited to generate unfair or harmful representations, potentially leading to discriminatory outcomes if deployed in sensitive applications.

Future work should prioritize the development of safeguards against such misuse. This could involve research into methods for detecting and rejecting biased or malicious instructions, establishing protocols for the responsible deployment of steerable vision models, and fostering a deeper understanding of the societal implications as this technology matures.

### A.4 SAFEGUARDS

Since our training set is repurposed from Imagenet dataset and other established benchmarks that has been extensively used by the field, they shall not contain image data with NSFW. For language instructions, one can implement a classifier for the instructions to classify if it is benign or malicious as a straightforward safeguard.

### A.5 PSEUDO CODE

We provide pseudo code for implementing our LIVE encoder and training loss.

```
1  # Assuming text_query, image, text_answer are input batches
2  # Assuming t (temperature) and b (bias) are parameters
3  # Models: text_query_model, image_model, text_answer_model
4
5  # Precomputed query:
6  _zquery_raw, out_query = text_query_model(text_query)
7  zquery = jax.lax.stop_gradient(_zquery_raw)
8
9  # Image embeddings steered by query:
10 zimg, out_img = image_model(image, query_tokens=zquery)
11
12 # Text answer embeddings:
13 _ztxt_raw, out_txt = text_answer_model(text_answer) # **kw omitted
14 ztxt = jax.lax.stop_gradient(_ztxt_raw)
15
16 # Compute logits:
17 logits = jnp.dot(zimg, ztxt.T) # (batch_size, batch_size)
18 logits = logits * t + b
19
20 # Contrastive loss calculation:
21 batch_size = zimg.shape[0]
22 eye = jnp.eye(batch_size)
23 m1_diag1 = -jnp.ones_like(logits) + (2 * eye)
24
25 loglik = jax.nn.log_sigmoid(m1_diag1 * logits)
26 nll = -jnp.sum(loglik, axis=-1) # NLL per sample
27 loss = jnp.mean(nll) # Average loss for the batch
```

Figure 8: Pseudo JAX code for language-steered vision embedding model.

```
1   # ViT Input: Image + Language Query Tokens (Concise)
2   # Assumes: self (Flax Module), nn (flax.linen), jnp (jax.numpy)
3   # Config: self.T, self.dtype_mm, self.width, self.patch_size, self.posemb
4   # Helper: get_posemb() for positional embeddings
5
6   # 1. Image to Patch Embeddings
7   img_in = jnp.asarray(image, dtype=self.dtype_mm)
8   patches = nn.Conv(features=self.width,
9                     kernel_size=(self.patch_size, self.patch_size),
10                    strides=(self.patch_size, self.patch_size),
11                    padding="VALID", name="patch_conv",
                      ↪  dtype=self.dtype_mm)(img_in)
12  n, h, w, c = patches.shape
13  patch_emb = jnp.reshape(patches, (n, h * w, c))
14  # Add positional embeddings to patch embeddings
15  patch_emb += get_posemb(self, self.posemb, (h,w), c, "patch_pos",
    ↪  patch_emb.dtype)
16
17  # 2. Process Query Tokens
18  # query_tokens input, e.g., (batch, query_feat_dim)
19  q_proj = nn.Dense(features=c * self.T, name="query_proj",
20                    dtype=self.dtype_mm)(query_tokens)
21  q_proj = jnp.reshape(q_proj, (n, self.T, c))
22  q_pos_emb = self.param("query_pos_emb", nn.initializers.zeros,
23                    (1, self.T, c), self.dtype_mm)
24  query_emb = q_proj + q_pos_emb
25
26  # 3. Concatenate query and patch embeddings for ViT Encoder
27  # Typically, sequence_axis=1 for (batch, seq_len, features)
28  encoder_input = jnp.concatenate([query_emb, patch_emb], axis=1)
29
30  # 'encoder_input' is then fed into the main ViT Encoder layers
```

Figure 9: Concise pseudo JAX code for ViT input processing with language queries. The $self.T$ is number of language tokens feed into the Vit.

### A.6 COMPARISON WITH EXISTING WORK

We list a comparison with existing vision language models in the followings, and visualize their architecture in Figure 10.

- A) CLIP Radford et al. (2021), SigLip Zhai et al. (2023), LiT Zhai et al. (2022b)
- B) Llava Liu et al. (2023), Gemma Team et al. (2025), Paligemma Beyer et al. (2024), Llama Grattafiori et al. (2024)
- C) CoCA Yu et al. (2022), Cappa Tschannen et al. (2023)
- D) VILT Kim et al. (2021)
- E) Falmingo Alayrac et al. (2022), BLIP Li et al. (2022), X-former Swetha et al. (2024)
- (F) Ours

Our work introduces the first vision-centric encoder that uses language to modulate visual computation for encoding target tasks. We address the scarcity of high-quality image, query, and answer triplet data by transferring the knowledge from LLM such as Gemini, and we demonstrate how language can directly control the vision encoder.

### A.7 THE IMPACT OF LANGUAGE INSTRUCTIONS FOR LIVE

Since our method allows feeding text instructions to the vision encoder, we have the potential to serve the final task better by improving the query. We investigated the impact of prompt text on the ImageNet classification accuracy of our SigLIP So400M model variant. We show the classification

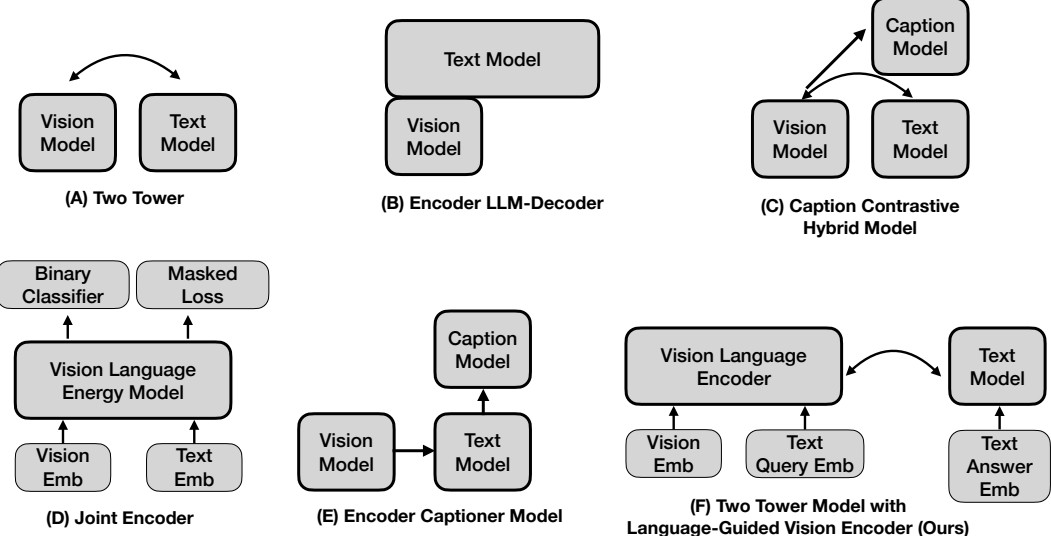

Figure 10: Comparison with existing methods. Note that B, C, E requires large language model based decoders. D does not have a embedding to perform zero-shot retrieval.

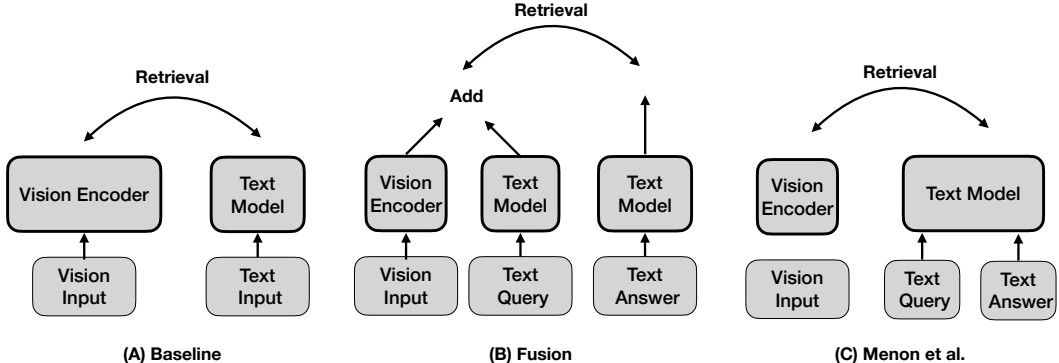

Figure 11: Illustration for baselines compared with in our paper. We take the two tower architecture (A), add the text query embedding to the embedding (B), and adding query to the text answer as description following Menon et al. (C).

accuracy of different prompt in Figure 12. Due to our model's training on more sophisticated image queries, the ImageNet classification accuracy dropped to 49.32% when no query prompt was used in retrieval tasks. Interestingly, by leveraging Gemini to evolve and generate different text prompts Yang et al. (2023), we improved the ImageNet accuracy to 68.18% using the instruction query: "Classify the main object." We believe this demonstrates the potential of our instructive vision foundation model for future work in prompt optimization to achieve even higher accuracy.

## A.8 ADDITIONAL EXPERIMENTAL RESULTS.

**Results on Steering Visual Representations for Text Recognition.** We repurposed the ImageNet dataset for a text recognition task by rendering text from one ImageNet category onto an image of another. A visual representation that ignores this text and instead predicts the original image's category would result in 0% accuracy. Therefore, higher accuracy directly indicates the model's ability to follow instructions and perform OCR retrieval. As shown in Table 7, our approach demonstrates significant effectiveness.

**Robustness Against Typographical Attacks.** Vision-language models like CLIP and SigLip are known to be vulnerable to typographical attacks, where target text is appended to an image to mislead

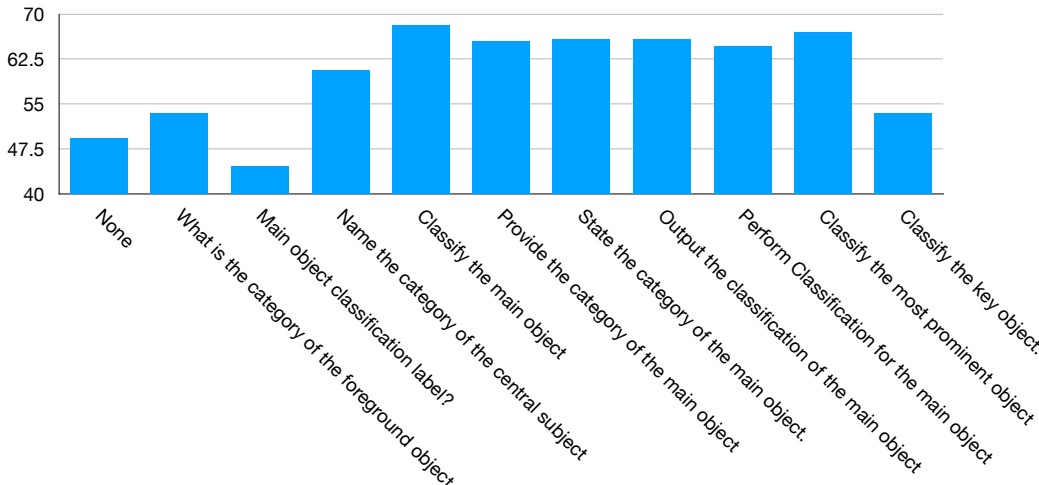

Figure 12: **Impact of Different Language Instructions for ImageNet classification task.** The y-axis shows the ImageNet classification accuracy in %. The x-axis shows the language instructions for the vision encoder. By improving the query prompts, we can improve the downstream task accuracy by up to 20 points.

| ViT Model | OCR Accuracy | |
| | Baseline | Ours |
| --- | --- | --- |
| SigLIP 2 ViT-SO-14 | 10.48 | **38.99** |

Table 7: Zero-shot accuracy to recognizing the text on Imagenet dataset. We evaluate OCR performance when text in words, potentially of different categories to the ImageNet image, is rendered in the image. If the model only perceive the original imagenet image without attending to the added text, the accuracy will be 0. While vision-only representations has a low accuracy on recognizing the text, our instructive visual embeddings allow embedding either image or text information based on instructions.

the model's representations. This vulnerability poses a significant concern for critical applications such as autonomous driving and facial authentication.

To evaluate this, we rendered a text sticker in the middle of ImageNet images, with the text explicitly stating a different class name than the original image. If the model attends to this text sticker, its accuracy drops to 0%. As shown in Table 8, baseline models exhibited a reduced ImageNet accuracy of 48.31% under these attacks. However, simply by adding the prompt, "Ignoring text, what is the object?", we observed a significant increase in robust accuracy, demonstrating our approach's ability to disregard typographical attacks.

| ViT Model | Robustness Against Typographical Attacks | |
| | Baseline | Ours |
| --- | --- | --- |
| SigLIP 2 ViT-SO-14 | 48.31 | **51.48** |

Table 8: **Zero-Shot test accuracy on ImageNet with typographical attacks.** When providing text sticker on top of the image, original image classification model has the tendency to be mislead by the text. By using text prompt to let the model to ignore the text, we can increase the robustness against typographical attack.

**Instructive Visual Benchmark on All Language Instructions.** In the main paper, we present the results on testing on unseen instructions, where we exclude all the language instructions that appear in the training. In Table 9, we also show the results on all instructions, which includes also language instructions that appear in the training. Our method consistently improves accuracy on the instructive

| ViT Model | ImageNet | | | | Caltech 101 | | | |
|---|---|---|---|---|---|---|---|---|
| | SigLip | Fusion | Menon et al. | Ours | SigLip | Fusion | Menon et al. | Ours |
| SigLip T/14 | 7.84 | 10.85 | 7.93 | **71.72** | 7.52 | 6.64 | 9.49 | **26.50** |
| SigLip B/16 | 8.45 | 9.35 | 8.17 | **83.51** | 8.83 | 7.56 | 12.3 | **38.74** |
| SigLip 2 B/16 | 9.29 | 10.91 | 9.50 | **84.54** | 8.18 | 8.05 | 14.8 | **37.12** |
| SigLip 2 So400m | 9.21 | 10.46 | 9.43 | **85.00** | 8.08 | 8.18 | 15.04 | **37.64** |

| ViT Model | SUN | | | | RefCOCO | | | |
|---|---|---|---|---|---|---|---|---|
| | SigLip | Fusion | Menon et al. | Ours | SigLip | Fusion | Menon et al. | Ours |
| SigLip T/14 | 6.13 | 5.50 | 6.72 | **26.87** | 5.72 | 5.72 | 5.72 | **33.52** |
| SigLip B/16 | 8.41 | 8.06 | 9.68 | **41.55** | 6.93 | 6.94 | 7.09 | **47.24** |
| SigLip 2 B/16 | 10.08 | 10.02 | 14.41 | **41.41** | 7.75 | 7.75 | 9.72 | **45.42** |
| SigLip 2 So400m | 10.81 | 10.54 | 15.26 | **44.68** | 6.63 | 6.68 | 7.65 | **47.80** |

Table 9: **Zero-Shot Accuracy on Instructive Visual Benchmark repurposed from ImageNet, Caltech 101, SUN, and RefCOCO.** We directly test our model on these datasets without any training on them. This is in contrast to prior work that require finetuning on those downstream tasks Mao et al. (2023); Beyer et al. (2024) to do them.

visual benchmark. Despite some instructions being encountered during training, the task's difficulty persists. This is attributed to the new image and data domains, and the fact that many tasks remain non-trivial even with instruction familiarity.

**Ablation Study on Cross-Instruction Generalization** We investigate the ability of our learned embeddings to generalize to unseen instruction families after training on a distinct set. Utilizing Gemini, we automatically categorize ImageNet instructions into four broad families: fundamental properties (F), spatial-textual symbolic tasks (S), viewpoint composition aesthetics tasks (V), and dynamic inferential interpretive reasoning tasks (D).

Table 4 presents our results where a SigLip 2 B/16 model is trained on three of these instruction groups and evaluated on the deliberately held-out fourth group on ImageNet. While training and testing on all groups yields 86.93% accuracy, testing on our hold-out subgroups results in only a 1-2 percentage point accuracy drop for three of our studies. Notably, when not training on F (fundamental properties), the model experiences a significant accuracy drop, underscoring the importance of training on instructions related to fundamental properties.

## A.9 TRAINING DATASET

We conducted an in-depth analysis to understand the distribution of language instructions generated by our LLM for the ImageNet dataset. Our process involved two key steps: First, we used Gemini Flash 2.0 to define 66 distinct subcategories for vision-related questions, which are depicted in Figure 13. Second, we employed Gemini Flash 2.0 to assign each question within our expansive 16-million synthetic image-query-answer triplet dataset to one of these 66 categories, or to an "others" category if it didn't fit.

The resulting distribution, visualized in Figure 13, reveals significant variations in instruction frequency. "Material identification via Visual Properties" was by far the most common, accounting for roughly 2.2 million data entries. In contrast, "Fractal Properties/Self-Similarity Analysis" was rarely observed, with only 140 associated queries.

## A.10 TESTING DATASET

### A.10.1 ESTABLISHED BENCHMARKS

**MMVP.** In this paper, we use the MMVP-VLM benchmark, which are divided into 9 visual patterns. The benchmark consists of image pairs with corresponding answer pairs to retrieve. The original MMVP only comes with text answers, no text queires. Yet since they are divided into 9 categories with answers that has a good description for the task to ask about. We create text queries, which itself,

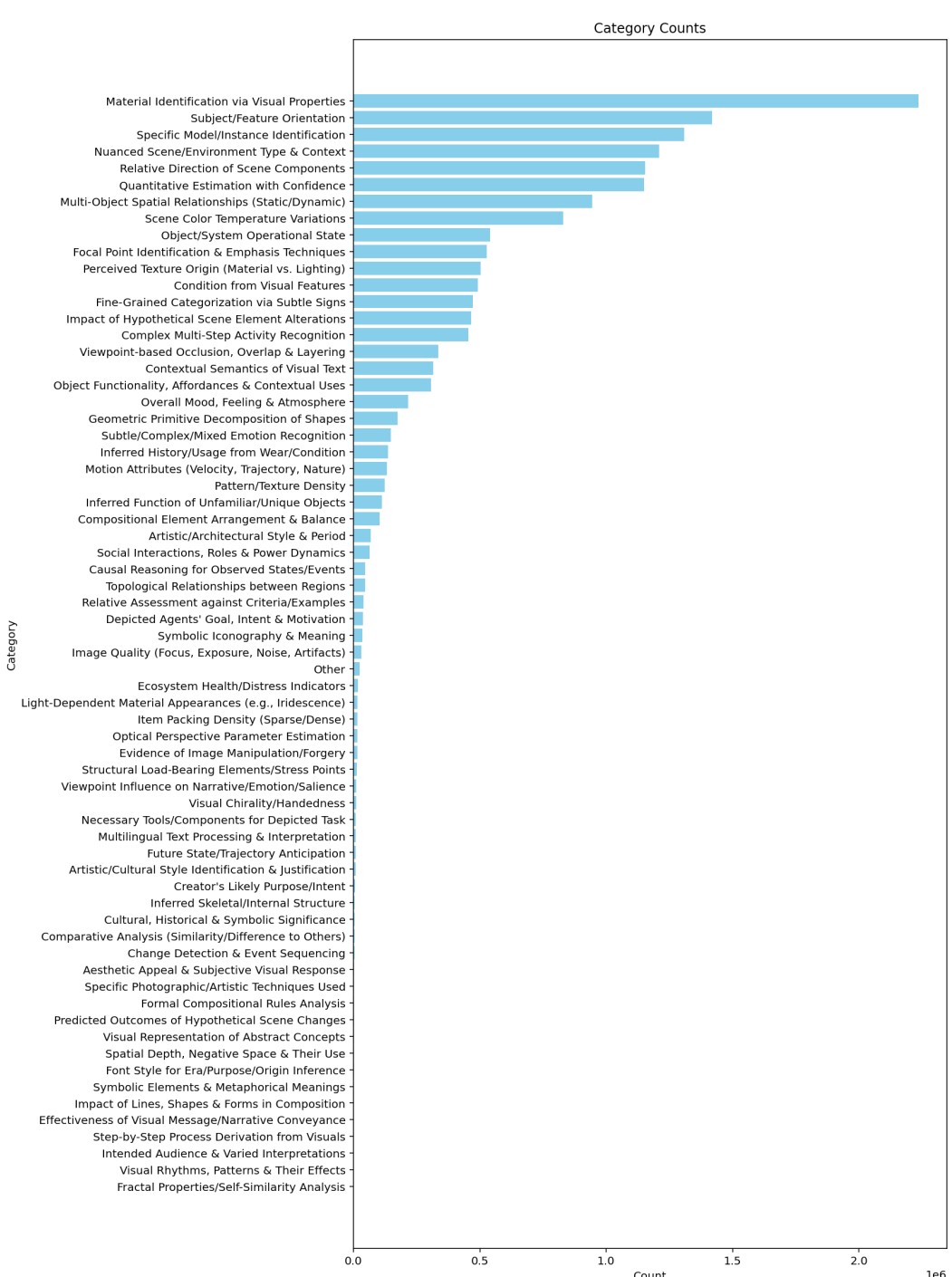

Figure 13: **The Histogram of Query Categories Generated in our Language-Instructed ImageNet.**
We first use LLM to generate a taskonomy of visual queries. We then use LLM to label each instructions we generate to one of the categories. We show the counting plot. The data generated show a long tail distribution.

does not offer any additional information to distinguish the text answer, since for example, the answer pair to discriminate is "A minion smiling with tounge out" and "A minion smiling without tounge out", our added query: "Is the minion smiling with tounge out" does not offer additional information

on the tounge's status, but a repeat of the answer context. Note the chance prediction accuracy on MMVP is 50% due to the binary choice.

**VQA v2, GQA.** While our target is to evaluate how language can steer the visual representations, as benchmarked by our above datasets that designed to evaluate this, like having rich query and answer pair for the same image. We also use existing visual question answering tasks like VQAv2 and GQA, which often has single query and ansewr for the same image. We subsample the first 500 samples from GQA to validate our approach for all our experiments. By adding our instructions, we achieve significantly higher accuracy than vanilla models. Note that for VQA task, the query tend to already contain a lot of information about the image, therefore, the Menon et al achieve higher accuracy largely due to the query allows better retrieval for the image.

### A.10.2 Our Benchmarks

For the following repurposed dataset to evaluate this language-instructed vision embeddings, we use the same prompt to generate the test answer and query:

```
Provide a numbered list of interesting visual questions about the image,
    followed by the corresponding answers.
```

Note since those are unseen images, and new domains for four of them, the Gemini-generate questions are often very different, allowing us to perform zero-shot evaluation on both: 1) novel data category and domain but instructions could be seen before, 2) novel data category, domain, and unseen instructions. We report the (2)'s results in the main paper due to the space limitation. We will also report the accuracy for (1) in later section.

**ImageNet.** ImageNet validation was originally designed for evaluating classification tasks. We repurpose it to also benchmark instructive visual embeddings. The queries are generated by gemini condition on the Image. In the main paper, we remove all instructions are appear in the training. Therefore, the numbers shown is on unforeseen, new instructions. In addition, in the appendix, we also show the retrieval on all the instructions generated without removing the ones that overlap with the training queries. There are 145549 data for the validation data in the paper after removing the ones with instructions appear in the training. Before removing the data with seeing instructions, is 551514. We retrieve the answer from 1000 answers, which contains the groundtruth and 999 random others.

**Caltech101** We repurpose the test set via Gemini, to generate open queries and corresponding answers. In the main paper we remove queries that overlap with training. We also show the results for the set without removing the overlapping ones.

**SUN** We repurpose the test set via Gemini, to generate open queries and corresponding answers. In the main paper we remove queries that overlap with training. We also show the results for the set without removing the overlapping ones.

**RefCOCO** We use the images with rendered bounding box to to create the test datasets. We feed the image with bounding box to LLM to generate open queries and corresponding answers. Note that the task is zero-shot because bounding box is not given in ImageNet.

**ImageNet OCR Test** We render text on the ImageNet validation images, where the text are the name of a different category. Therefore, the model will have different predictions by looking at the text or the image object category itself.

### A.11 Attention Visualizations

We provide more attention visualizations of our encoder in Figure 14. Guided by language instructions, without supervision on where the model shall look at, our LIVE encoder learns to focus on the part of the image that is corresponding to the language instructions.

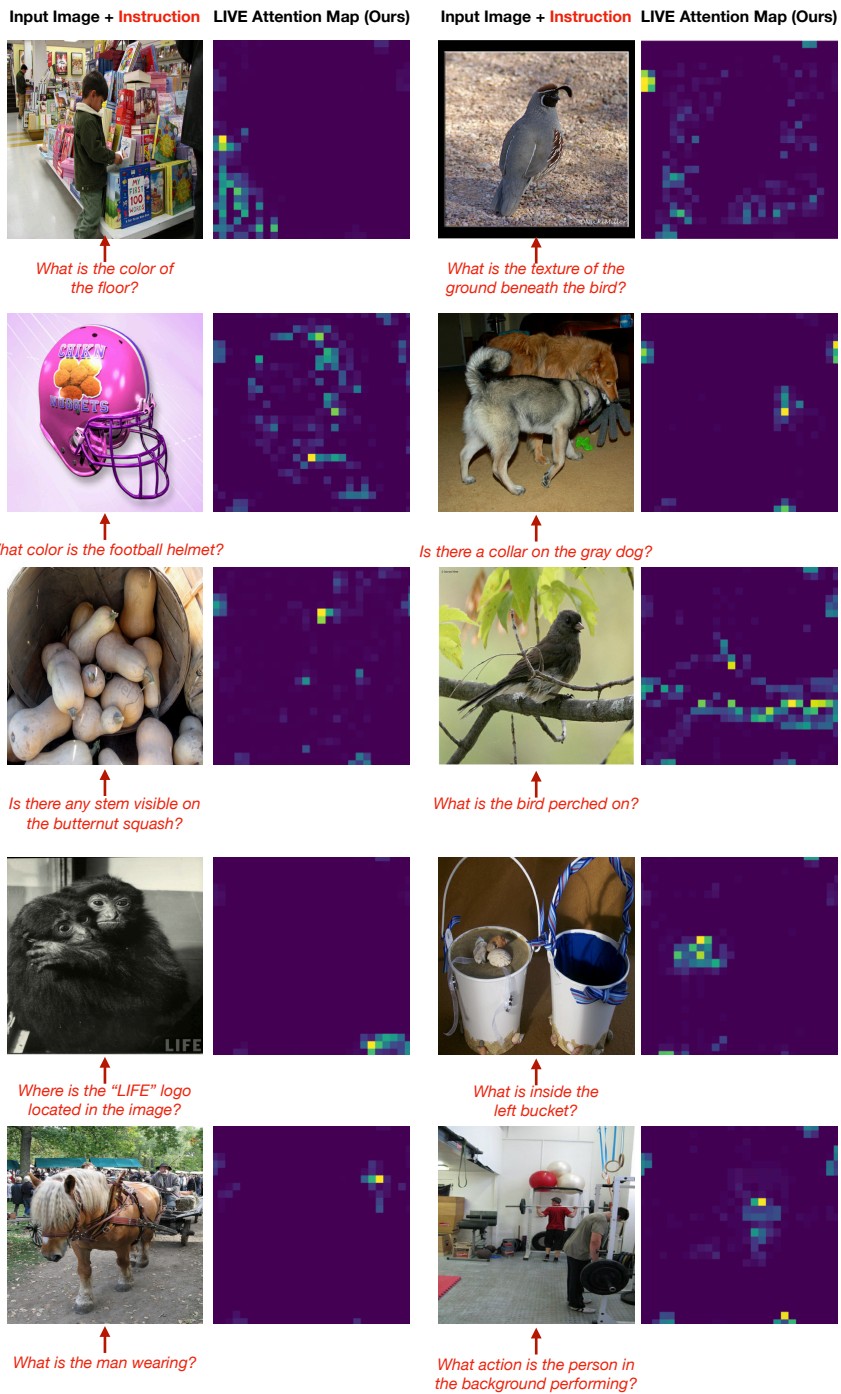

Figure 14: **Attention Visualizations of Our LIVE Encoder.** Guided by language instructions, the ViT model learn to focus on relevant parts, effectively prioritizing information and ignoring distractions. This is achieved without any direct supervision on the region the model shall focus on, showing the active, selective capabilities can be automatically learned by our encoder. Examples are randomly draw from ImageNet validation set that was not trained on.

