# OpenReview forum: "Language-Instructed Vision Embeddings for Controllable and Generalizable Perception"
_ICLR.cc/2026/Conference — ICLR 2026 Poster_

### Official Review · Reviewer_mP9X · 2025-10-28

**Soundness:** 3
**Presentation:** 3
**Contribution:** 3
**Rating:** 8
**Confidence:** 3

**Summary:**

This paper proposes Language-Instructed Vision Embeddings (LIVE), a novel paradigm that uses language instructions to dynamically guide a vision encoder during inference, enabling it to produce task-specific visual embeddings without retraining. To achieve this, LIVE reuses the text encoder from SigLIP and injects textual queries directly into the vision part, allowing it to focus on instruction-relevant image regions. After training on  around 16.4 million images-query-answer synthesized by Gemini, LIVE reduces visual hallucinations by 34 points on the MMVP benchmark and outperforms much larger vision–language models (e.g., LLaVA, InstructBLIP) on GQA, demonstrating its effectiveness.

**Strengths:**

1. The idea of this paper is quite novel. By training a language-instructed visual embedding model, it addresses visual tasks more efficiently with a model of reduced parameter count, eliminating the need for task-specific training. Furthermore, it mitigates the issue of large language models (LLMs) dominating visual inference in existing multimodal large language models (MLLMs), which is a commendable contribution.
2. The methodology is both straightforward and effective. Moreover, the synthesized triplet dataset holds significant potential for enhancing CLIP-style pretraining for the research community.
3. The experimental results are promising, demonstrating that a considerably smaller embedding model surpasses existing MLLMs across a wide range of tasks. Additionally, the attention visualizations substantiate the efficacy of LIVE at the representational level, which is highly appreciated.

**Weaknesses:**

1. Generating large-scale vision-related data based on Gemini 2.0 may incur substantial costs. It would be beneficial if the authors could provide relevant details regarding the expenses associated with data synthesis and model training.
2. The examples presented in the paper are confined to relatively simple queries. It remains an open question whether the proposed approach maintains its efficacy when applied to more complex visual problems and intricate queries (e.g., visual reasoning tasks).

**Questions:**

1. Can the trained visual encoder be used for MLLM training? Will it make current MLLM (.e.g. LLaVA) better or worse?
2. See Weakness above

---

> ### Author Response · Authors · 2025-12-03
> **Thank you for your review, we addressed the questions below**
>
> We thank Reviewer mP9X for the positive review. We are glad that the reviewer recognized the core novelty and contributions of our work, such as its efficiency, the reduced parameter count, and its role in "mitigating the issue of large language models (LLMs) dominating visual inference... a commendable contribution." We sincerely appreciate this strong support and "accept" recommendation.
>
> We are happy to provide details on the points raised.
>
>
> **W1: Cost of Data Synthesis and Training**
>
> 1. Data Synthesis Expenses: We utilized Gemini 2.0 Flash for annotation due to its high cost-efficiency ($0.10/1M input tokens, $0.400/1M output tokens). The total one-time cost to generate 16.4M triplets from the 1.2M ImageNet images was approximately $1,000 USD. To eliminate this barrier for the community, we will publicly release the complete dataset upon publication.
>
> 2. Training Resources: For model training, we utilized up to 256 TPUv3 cores. Given the efficiency of our model (only 0.9B parameters) compared to large multimodal baselines, the training cost is standard for vision-language pre-training and reproducible in academic settings with moderate compute resources.
>
> **W2: Efficacy on Complex Visual Reasoning**
>
> 1. Quantitative Evidence (GQA):
> We see early evidence that our model can perform complex queries. We evaluate on GQA, a benchmark specifically designed to test compositional visual reasoning involving spatial relations, logic, and multi-step attribute filtering.
> Result: LIVE achieves 67.4% accuracy on GQA.
> Implication: This strong performance confirms that the model handles structural complexity (e.g., "Is the bowl to the left of the glass full?") effectively, going significantly beyond simple object recognition.
>
> 2. Qualitative Evidence (Multi-Hop Chains):
> As shown in Figure 5(4), the model successfully executes multi-step spatial reasoning instructions: "What color is the motorcycle to the right of the motorcycle highlighted by the bounding box?"
> This requires a sequential chain: (1) Locate Box $\to$ (2) Apply Spatial Relation ("right of") $\to$ (3) Identify Target $\to$ (4) Extract Color. The success on these inputs demonstrates that the "Language $\to$ Vision" paradigm supports complex, serial visual processing without requiring a full LLM decoder.
>
> **Q1: Usability in MLLM Training (e.g., LLaVA)**
>
> This is an excellent suggestion and a very exciting avenue for future research.
> Feasibility: Yes, the LIVE encoder can theoretically replace the standard CLIP backbone in architectures like LLaVA. This would require a slight modification to the LLaVA architecture to feed the text prompt into the vision encoder (Language $\to$ Vision) before the features are passed to the LLM.
>
> Better or Worse?:
> We hypothesize that using LIVE would improve MLLM performance, particularly on reasoning tasks, due to "pre-computation":
> * Standard LLaVA: The LLM receives a static, global visual representation (CLIP) and must perform all attention/filtering internally.
> * LIVE-LLaVA: The vision encoder would act as a dynamic filter, passing features that are already "instruction-aware" to the LLM.
>
> By increasing the signal-to-noise ratio of the visual tokens before they reach the projector, this could reduce the reasoning burden on the LLM and mitigate hallucinations.

---

### Official Review · Reviewer_k7wj · 2025-10-31

**Soundness:** 3
**Presentation:** 3
**Contribution:** 2
**Rating:** 4
**Confidence:** 4

**Summary:**

This paper aims to make vision encoders, such as CLIP, to extract language-instructed vision embeddings to produce query-specific visual features for downstream applications. To address this, the authors let Gemini-2.0-Flash annotate over 16M image-question-answer pairs, and then fine-tune the original vision encoder with SigLIP's loss. Experiments under various evaluation protocols demonstrate the effectiveness of the proposed method.

**Strengths:**

1. This paper is well-written and easy to follow.
2. The motivation is clear, reasonable, and meaningful.
3. The proposed method is quite effective.

**Weaknesses:**

1. The improvement may mainly come from the data instead of the method. To investigate this, the following experiments are highly recommended:
   - Train LIVE on the classic caption dataset (using "describe this image" as the prompt) or the original ImageNet dataset (using "what is the category of this image" as the prompt).
   - Train other methods like Flamingo or BLIP-2 using the proposed data.

2. How this encoder incorporates with typical MLLMs remains unclear. It is encouraged to replace the original CLIP-336 in LLaVA with LIVE.

**Questions:**

N/A

---

> ### Author Response · Authors · 2025-12-03
> **Thank you for your review, we addressed the questions below.**
>
> We sincerely thank Reviewer k7wj for acknowledging that our paper is "well-written," "easy to follow," our motivation is "clear, reasonable, and meaningful," and our method is "quite effective." We address the reviewer’s questions below:
>
>
> **W1: Isolating Method Improvements from Data**
>
> 1. Experiment A: Train LIVE on Standard/Classic Data
> To demonstrate that the architecture specifically requires our instructional logic (and cannot simply "work" with standard data), we performed the requested ablation. We retrained the LIVE model using:
>
> * Neutral Captions: Replacing specific queries with "describe this image" (similar to CC3M/WebLi).
> * Standard Classification: Replacing rich answers with ImageNet class names (e.g., "What is the category?").
>
> Result: As shown below, performance collapses without the specific instruction-answer synergy. The method requires the Language $\to$ Vision instruction flow; it does not passively benefit from standard labels.
>
> | Experiment | Text Query| Text Target | GQA (Reasoning) | MMVP (Perception) |
> | :--- | :--- | :--- | :--- | :--- |
> | LIVE (Full) | Ours | Specific Query | 67.4 | 69.5 |
> | Ablation 1 | describe this image | Neutral Prompt | 13.1 | 65.1 |
> | Ablation 2 | what is the category of this image | Class Name | 2.7 | 54.7 |
>
> 2. Experiment B: Train Flamingo/BLIP-2 on Proposed Data We did not perform this experiment due to a fundamental difference in scope and computational regime.
>
> * Efficiency Mismatch: LIVE is designed for efficient representation learning (~0.9B parameters) without an LLM. Flamingo (80B) and BLIP-2 (14B) operate in a massive parameter regime.
> * Goal: Our contribution is demonstrating that a compact vision encoder can achieve reasoning capabilities without scaling to LLM sizes. Comparing against or training 80B-parameter models would obscure this specific efficiency contribution.
>
>
> **W2: Integration with MLLMs (LLaVA)**
>
> 1. Architectural Mismatch:
> We clarified that replacing CLIP in LLaVA with LIVE is not merely an engineering swap but a contradiction of our core paradigm.
> LLaVA (Vision $\to$ Language): Relies on a massive LLM decoder to perform reasoning after visual encoding.
> LIVE (Language $\to$ Vision): Moves the reasoning burden into the encoder itself using instruction conditioning.
> Using LIVE merely as a static feature extractor for LLaVA nullifies the "steering" mechanism that drives our performance.
>
> 2. Efficiency Argument:
> Our primary contribution is proving that visual reasoning is possible without a heavy LLM decoder.
> LIVE: <1B parameters (Standalone).
> LLaVA: 7B+ parameters.
> Plugging LIVE back into LLaVA reintroduces the massive computational overhead we specifically aimed to eliminate. Thus, we position LIVE as an efficient, standalone alternative to MLLMs, not a plugin module for them.

---

### Official Review · Reviewer_pP9q · 2025-11-01

**Soundness:** 2
**Presentation:** 3
**Contribution:** 3
**Rating:** 4
**Confidence:** 4

**Summary:**

This work proposes a language-instructed vision embedding (LIVE)  method, aiming at extract more controllable and generalizable visual representation upon the language guidance. Specifically, the vision encoder takes the embedding of the textual instruction q and the image as input and compute the instructive visual embedding. Extensive experimental results compared with a series of the leading methods show the improved performance of the proposed method.

**Strengths:**

* The proposed method proposes a new perspective of learning effective and adaptive visual representation, which is interesting.

* This work is well-motivated.

* The introduction is well-written.

**Weaknesses:**

* L156~160 demonstrate that the vision encoder always requires the query as the instruction to guide the visual representation learning. Although the concept is novel and could make sense, this strategy inevitably brings the bypass for the multimodal learning, e.g., model can   rely on the query, rather than the image itself to learn the representation for the semantic matching. Intuitively, this might be one of the cause for the performance boost.

* During training, the answer is adopted. Yet, during inference, it seems that the answer may not be adopted. Is this mismatch intentionally devised and what might be the rational behind this?

 * This work provides extensive of performance comparison but lacks thorough ablation and model discussions in the manuscript, such as the discussion of each type of the input to the text/img encoders.

* Fig.6 shows that the query steer the visual attention. Yet, the more effective the steering it is, the more important role the query plays for the representation learning, instead of visual signals.

**Questions:**

* What is the setting of the inference pipeline? What are the specific inputs of the visual encoder and the text encoder? More detailed illustrations are expected.

* The proposed method highlights more on the semantics of the visual embedding, how's the sensitivity of the proposed method to the visual detail variation?

* Why the proposed work is termed as "generalizable perception"? How does the proposed method relate and contribute to the visual perception and low-level patterns?

---

> ### Author Response · Authors · 2025-12-03
> **Thank you for your review, we addressed your questions below**
>
> We sincerely thank Reviewer pP9q for their time and for acknowledging that our work presents an "interesting" new perspective, is "well-motivated," and "well-written". We address the reviewer’s questions below:
>
> **W1: Concern regarding text shortcuts (bypassing visual learning).**
> We empirically rule out the possibility of the model relying on text shortcuts (ignoring the image) via our results on the MMVP benchmark, which is specifically designed to penalize text-only priors.
>
> 1. Structural Impossibility of Shortcuts: MMVP utilizes "minimal pairs"—images that differ by a single visual detail (e.g., "minion with tongue out" vs. "minion without tongue out") while sharing identical text contexts. In this setting, the query is symmetrically uninformative; it is semantically equidistant to both options. Logic derived solely from the text query cannot resolve the ambiguity.
>
> 2. Empirical Evidence: If LIVE were bypassing the visual input to rely on the query, performance on these pairs would collapse to random guessing. Instead, we achieve a +34 points accuracy gain over baselines. This massive improvement confirms that the query is not bypassing the image, but rather successfully guiding the encoder to extract the specific fine-grained visual features required to distinguish the minimal pairs.
>
>
> **W2: Logic behind Train/Inference "Mismatch".**
> We clarify that this is not a mismatch, but the standard contrastive learning formulation (identical to CLIP or SigLIP). The "answer" acts as the supervision target, not an input feature.
> * Training (Contrastive Alignment): The answer is used solely to define the ground-truth target for the loss function. The model optimizes the (Image + Query) embedding to align with the correct Answer embedding while pushing away negative answers in the batch.
> * Inference (Retrieval/Selection): The answer is naturally not an input. The trained model generates an embedding for the (Image + Query) which is then used to rank/retrieve the closest match from a bank of candidate answers.
> This protocol—training on ground-truth alignment and inferring via nearest-neighbor retrieval—is the standard evaluation setup for discriminative models on benchmarks like GQA and MMVP.
>
> **W3: Ablation Studies on Input Types (Vision & Text)**
> We deepened our analysis by isolating the impact of the specific input types used for both the vision and text encoders.
>
> 1. Vision Encoder Input (Instruction Specificity):
> We replaced our task-specific queries with a single neutral prompt (e.g., "describe this image") to test if the content of the instruction matters.
> Result: Performance on reasoning tasks (GQA) collapses from 67.4 $\to$ 13.1.
> Insight: This confirms that the vision encoder requires specific, query-driven steering to extract relevant features; a generic "global" instruction is insufficient for reasoning.
>
> 2. Text Encoder Input (Supervision Granularity):
> We replaced our pipeline-generated descriptive answers with standard ImageNet class names (e.g., Target: "Goldfish") to test if standard categorical supervision is sufficient.
> Result: Performance collapses to near-random guessing on GQA (2.7).
> Insight: Standard classification labels (nouns) are insufficient for learning visual relationships. The rich, descriptive structure of our generated answers is the key driver of the model's reasoning capability.
>
> | Method | Vision Input | Text Target | GQA (Reasoning) | MMVP (Perception) |
> | :--- | :--- | :--- | :--- | :--- |
> | Ablation 1 | Neutral Prompt | Rich Answer | 13.1 | 65.1 |
> | Ablation 2 | Specific Query | Class Name | 2.7 | 54.7 |
> | LIVE (Ours) | Specific Query | Rich Answer | 67.4 | 69.5 |
>
> **W4 Query Steering vs. Visual Signal Importance**
>
> 1. Mechanism (Steering $\neq$ Replacement):
> We clarify that effective steering implies visual focusing, not visual replacement. As shown in Figure 6, the query acts as a conditional attention mechanism. It does not allow the model to ignore the image; rather, it directs the visual encoder to amplify specific spatial regions (e.g., the "tongue of minion" ) while suppressing irrelevant background noise. This increases the signal-to-noise ratio of the visual features, making the visual representation more precise, not less important.
>
> 2. Empirical Proof (MMVP):
> If strong steering meant the model was relying on the query instead of the visual signal, performance on MMVP would collapse. As detailed in W1, MMVP queries are symmetrically uninformative (the text is identical for both image choices). The query contains zero discriminative power on its own. Our +34 points gain on this benchmark empirically proves that the query is not bypassing the image, but is successfully guiding the encoder to extract the specific visual evidence required to resolve the ambiguity.

---

> > ### Author Response · Authors · 2025-12-03
> > **Continued**
> >
> > Answers to Questions
> >
> > **Q1: What is the setting of the inference pipeline?**
> >
> > The pipeline is a standard retrieval/matching process:
> >
> > Input: The model receives an Image and a text Query, all candidate answers are provided.
> >
> > Step 1: The Image and Query are fed into the Vision Encoder (LIVE) to produce a single, instruction-aware visual embedding, $v_q$.
> >
> > Step 2: All candidate answers (e.g., all answers in the GQA test bank, or the two "minimal pair" answers in MMVP) are fed through the Text Encoder to produce text embeddings, $t_1, t_2, ..., t_n$.
> >
> > Step 3: We compute the cosine similarity between $v_q$ and all candidate text embeddings.
> >
> > Output: The model's "answer" is the text candidate with the highest similarity (e.g., $\text{argmax}_i (\text{sim}(v_q, t_i))$).
> >
> >
> > **Q2: Sensitivity to Visual Detail Variation**
> >
> > We empirically demonstrate that our method’s semantic focus enhances, rather than reduces, sensitivity to visual detail. The MMVP benchmark is explicitly designed to test robustness to fine-grained visual variations (e.g., distinguishing "a minion with tongue out" vs. "without tongue out"). Standard "semantic" models (like CLIP) often fail here because they encode generic features and are burdened to also encode the visual detail variation. Our +34-point gain on this benchmark is direct, empirical proof that our method dramatically improves sensitivity to these exact visual details.
> >
> >
> > **Q3:"Generalizable Perception" and Low-Level Patterns**
> >
> > Rationale for "Generalizable Perception": We use this term to describe the shift from static feature extraction to adaptive computation.
> > * Static (Standard): A standard encoder produces a fixed representation for an image, regardless of the downstream task.
> > * Generalizable (Ours): LIVE treats the instruction as a dynamic filter, allowing the same vision backbone to generate distinct feature maps for the same image depending on the query. This allows the model to generalize to unseen tasks at test time without retraining, simply by changing the prompt.
> >
> > Contribution to Low-Level Perception: Our method contributes to visual perception by demonstrating that language can effectively steer low-level visual primitives, not just high-level semantics.
> > * Evidence: We point to our performance on MMVP sub-categories which isolate fundamental, low-level patterns. We achieve massive gains over baselines on low-level pattern subtasks like:
> > Orientation/Viewpoint: +53 points (e.g., distinguishing left vs. right facing).
> > Counting: +40 points.
> > Color/Texture: +20 points.
> > * Mechanism: This confirms that the instruction successfully reconfigures the encoder's attention to resolve fine-grained spatial and textural ambiguities that standard semantic encoders (like CLIP) abstract away.

---

### Official Review · Reviewer_LBmL · 2025-11-03

**Soundness:** 2
**Presentation:** 3
**Contribution:** 3
**Rating:** 4
**Confidence:** 4

**Summary:**

The paper introduces a new paradigm in vision–language modelling: rather than merely feeding image features into a large language model, they reverse this flow by using language instructions to guide the vision encoder. The proposed method, called Language-Instructed Vision Embeddings (LIVE), uses natural-language directives at inference time to steer the vision encoder into producing task-centric visual embeddings without any task-specific retraining.
In practice, this means the model can dynamically focus on aspects of an image relevant to the given instruction, resulting in more controllable and generalizable representations. Empirical results show substantial reinforcement: LIVE reduces visual “hallucinations” by a large margin and outperforms much larger vision–language models on visual question-answering tasks, even when faced with unseen instructions and tasks.

Contributions:
- Introduces the idea of language-driven visual encoding: language is not just describing or querying, but actively guiding how the visual encoder processes the image.
- Provides a framework where the vision encoder produces embeddings that are instruction-aware, enabling strong controllability (you decide what to focus on) and generalizability (works for new instructions/tasks).
- Demonstrates empirically that this paradigm can deliver strong performance gains: better alignment of visual features with linguistic task goals, fewer hallucinations, and robustness to unseen tasks/instructions.

**Strengths:**

The paper presents a well-motivated and timely contribution in the field of language-guided visual representation learning. The proposed LIVE (Language-Instructed Visual Encoder) introduces an instruction-based training paradigm that allows natural language to directly modulate the visual encoding process. This idea aligns closely with the current trend of multimodal alignment and reasoning, yet it explores an under-investigated direction—embedding-level linguistic control—rather than the more common dual-encoder or caption-generation settings.

Originality
The originality of LIVE lies not merely in architectural novelty, but in its reformulation of multimodal learning as instruction-conditioned representation alignment. Instead of aligning paired modalities post hoc (as in CLIP), the model internalizes linguistic guidance into the feature extraction process. This perspective opens new possibilities for controllable and interpretable vision encoders. While the high-level idea is inspired by prior “visual instruction tuning” work, the paper provides a conceptual synthesis that generalizes those methods into a unified embedding-learning view.

Quality
The paper provides a coherent training objective, a clear architecture description, and a reasonable learning pipeline involving LLM-generated image–instruction–answer triplets. The experimental section covers multiple benchmarks, showing consistent improvements over baseline vision encoders. The methodology is easy to follow, and the ablation studies—though limited—support the claim that linguistic instruction improves cross-task generalization. In terms of engineering quality, the model is computationally efficient and can be integrated into existing multimodal pipelines.

Clarity
The manuscript is generally well-written and clearly organized. Figures and tables effectively convey the core architecture and comparative results. The motivation is explicitly stated, and the connection to prior work is well articulated.

Significance
LIVE addresses a fundamental question in representation learning: Can visual features be directly instructed through language? The proposed framework provides early empirical evidence that such an approach is not only feasible but also beneficial. This work could inspire a new class of “instruction-aware” encoders applicable to downstream tasks such as retrieval, visual reasoning, and grounding. Its conceptual significance lies in bridging the gap between vision-language alignment and language-conditioned representation control, which is of high relevance to the ICLR community’s focus on generalizable and interpretable AI systems.

**Weaknesses:**

1.	Originality is under-argued vs. closely related paradigms.
The paper positions LIVE as “language-instructed” encoding inside the vision tower, but the boundary from prior lines of work (caption-conditioned encoder modulation, cross-attention/adapters, visual instruction tuning) isn’t made watertight.

2. Potential data contamination between training and evaluation remains insufficiently addressed.
Although the paper specifies that the 16.4 M (image, instruction, answer) triplets are generated with Gemini 2.0 Flash and even provides example prompts, it does not report any train–test decontamination analysis. Given that Gemini-generated supervision and evaluation “oracle” labels (marked †) are both derived from similar prompting pipelines, there is a tangible risk of semantic overlap or leakage into benchmarks.

3.	Use of a model-generated “oracle” undermines evaluation rigor.
Several benchmarks marked with “†” use Gemini-produced answers and then measure how close LIVE gets to that oracle. This is not a neutral ground truth: it bakes the teacher’s biases into the metric and can inflate apparent progress. Replace “oracle” labels with human annotations or established datasets; if teacher labels must be used, include teacher–student agreement vs. human gold, plus audits for leading prompts and spurious shortcuts.

4.	Fairness of headline gains is only partially established (scale mismatches remain).
The reported +34-point MMVP and large GQA gains are impressive. However, the overall training scale differs substantially—LIVE uses 16.4 M LLM-generated triplets trained on 256 TPUv3 cores—which makes it difficult to disentangle architectural improvements from data or compute advantages. Re-evaluating baselines under comparable compute/data budgets or reporting normalized gains would clarify how much of the improvement stems from the proposed mechanism itself.

5.	Ablations only partially support the claimed mechanism.
The paper provides some evidence that language modulates visual representations—such as comparisons without instructions, attention visualizations, and encoder-size studies—but these analyses remain high-level. More controlled ablations (e.g., varying where or how language is injected, or replacing instructions with neutral text) could strengthen the causal link between “instruction conditioning” and the observed performance gains. As written, it remains difficult to fully disentangle the benefits of language-guided encoding from those of larger data or improved contrastive training.

**Questions:**

1.	Clarifying Conceptual Novelty
	•	What exactly is unique about LIVE’s modulation pathway (e.g., position of language injection, training objective) that cannot be reduced to prior adapter or cross-attention methods?

2. Data Contamination (Weakness #2)
	•	Did you perform any explicit decontamination between the Gemini-generated triplets and the evaluation benchmarks?
	•	Would you consider running a controlled experiment on a held-out, human-annotated benchmark to quantify the possible inflation due to data leakage?

3.	Use of Model-Generated “Oracle” Labels
	•	For the “†” benchmarks where Gemini provides the reference answers, can you clarify the rationale for treating Gemini outputs as ground truth?
	•	How correlated are Gemini’s responses with human-labeled gold data (e.g., on a small manually annotated subset)?

4.	Scale and Fairness of Comparisons
	•	Can you provide more detail on the total compute budget and data volume of LIVE versus each baseline?
	•	Were any baselines retrained under identical compute/data conditions, or were all numbers taken from prior publications?

---

> ### Author Response · Authors · 2025-12-03
> **Thank you for your review, we address the questions below.**
>
> We sincerely thank Reviewer LBmL for their detailed feedback. We are glad that the reviewer recognized our work’s exploration of an "under-investigated direction—embedding-level linguistic control," and its potential to "open new possibilities for controllable and interpretable vision encoders." We appreciate the acknowledgment that our work "addresses a fundamental question in representation learning" and provides clear "empirical evidence that such an approach is not only feasible but also beneficial."
> We address the specific weaknesses and questions below.
>
> **W1: Clarifying Conceptual Novelty vs. Prior Paradigms**
> We appreciate the opportunity to clarify the boundaries of our contribution. Our core novelty lies not in the architectural components themselves, but in the reversal of the standard information flow—shifting from "Vision to Language" to "Language to Vision."
>
> * Comparison with Caption-Conditioned Modulation: While prior caption-conditioned methods (e.g., Lavoie et al., 2024,  Xiao et al., 2025) modulate the image using a single static caption, LIVE introduces a fundamental shift toward adaptive computation by utilizing multiple query-answer pairs for each image. This data structure allows the vision encoder to learn to dynamically extract different features for the same image depending on the specific query, rather than learning a single global representation.
>
> * Comparison with Cross-Attention/Adapters:* Our approach differs from standard cross-attention or adapter methods by intentionally preserving the standard ViT architecture via direct input token appending; this "minimal-change" design isolates our contribution to the data pipeline rather than architectural complexity.
>
> * Comparison with Visual Instruction Tuning: Finally, unlike standard visual instruction tuning (e.g., LLaVA) which relies on massive LLM decoders, LIVE injects language instructions directly into the vision encoder, achieving language-steered visual prediction with <10% of the parameters and eliminating the need for an LLM during inference.
>
> **W2: Clarification on Potential Data Contamination**
> We address the contamination concern by explicitly distinguishing between our primary evaluation benchmarks and our diagnostic metrics.
> * Independent Sources for Primary Benchmarks:
> Our core results are reported on human-annotated benchmarks (MMVP and GQA, as discussed in L311-L315). Since the training data is machine-generated (Gemini) while these evaluation sets are human-labeled, the data sources are independent by definition. There is no information leakage. The significant performance gains on these standard benchmarks (+34% on MMVP) confirm that the model has learned generalizable visual reasoning rather than memorizing generation artifacts.
> * Purpose of $\dagger$ Metrics:
> Regarding the datasets marked with ($\dagger$), we acknowledge that using the term "oracle" was confusing and have removed it in the revision. We clarify that these metrics are not intended to evaluate ground-truth task performance, but strictly to measure the knowledge transfer rate (distillation efficiency) from the teacher (Gemini) to the student (LIVE). These serve solely as a diagnostic tool to verify how well the student model aligns with its teacher. We add additional clarification sentences in L320.
>
> **W3: Clarifying "Oracle" Metrics vs. Human Ground Truth**
>
> We address the concern regarding evaluation rigor by clearly distinguishing the purpose of the two metric types used in the paper.
> 1. Correction on Terminology:
> We agree the term "oracle" was misleading and have removed it from the revision (L320, L390).
>
> 2. Distinct Evaluation Roles:
> * $\dagger$ Metrics (Diagnostic Only): These metrics are strictly used to measure the distillation efficiency—quantifying how much of the teacher's (Gemini) reasoning logic was successfully transferred to the student (LIVE). They are not used to benchmark task accuracy.
> * Human Benchmarks (Performance Claims): For all comparisons against SOTA and claims of absolute performance, we rely exclusively on established, human-annotated benchmarks (GQA and MMVP). Because these evaluation sets are authored by humans and disjoint from our training pipeline, they provide a neutral ground truth free from teacher bias.

---

> > ### Author Response · Authors · 2025-12-03
> > **More reply**
> >
> > **W4: Fairness of Comparisons (Scale & Data)**
> >
> > 1. Efficiency vs. SOTA: We emphasize that the scale mismatch puts us at a significant disadvantage, highlighting the efficiency of our method.
> > * BRAVE (SOTA): Uses 10B+ parameters and Billions of training images (LAION-2B, WebLi).
> > * LIVE (Ours): Uses 0.9B parameters and only 16.4M training triplets. Outperforming a model trained on orders of magnitude more data (+34.3% on MMVP) demonstrates that our gains stem from the instruction-guided mechanism, not raw scale.
> >
> > 2. Controlled Ablation (Isolating the Method): To confirm the performance is not due to the backbone (SigLIP) or the dataset existence, we trained a standard baseline ("Neutral Text") using the exact same backbone and 16.4M data, and use ViT-B model.
> > Result: Without the LIVE adaptive mechanism, the baseline collapses on reasoning tasks (GQA 13.1 vs. 67.4). This proves the data alone is insufficient; the instruction-guided architecture is required to unlock reasoning capabilities.
> >
> > **W5: Additional Ablation Studies (Mechanism Validation)**
> > On "varying where or how language is injected":  We compared injecting language tokens at different depths (Layer 1, 4, 8) of the ViT-B encoder.
> > Result: Early injection (Layer 1) yields the highest performance on MMVP (69.5), which requires preserving fine-grained visual details to detect hallucinations. Late injection (Layer 8) performs slightly better on GQA (68.2), likely because relation-focused tasks benefit from higher-level semantic abstraction.
> > Conclusion: This confirms that the language tokens actively modulate the visual feature hierarchy, rather than acting as a passive append.
> >
> > | ViT-B | GQA | MMVP |
> > | :--- | :--- | :--- |
> > | Layer 1 | 67.4 | 69.5 |
> > | Layer 4 | 67.8 | 69.4 |
> > | Layer 8 | 68.2 | 68.7 |
> >
> > 2.Neutral Text vs. Instructions (How language acts): To prove the "instruction conditioning" is the cause of the performance gain (rather than data scale or training duration), we replaced the specific query with a generic neutral prompt ("Caption the image") while keeping all other variables (backbone, data size, compute) constant.
> > Result: The model trained with neutral text completely fails on complex reasoning (GQA 13.1), whereas the instruction-guided model achieves 67.4.
> > Conclusion: This provides a strong causal link: the performance is not inherent to the data/backbone, but is caused by the specific instruction steering the encoder.
> >
> > | Method | Instruction Type | GQA | MMVP |
> > | :--- | :--- | :--- | :--- |
> > | Neutral Text: “Caption the image” (ViT-B) | Neutral “caption the image” | 13.1 | 65.1 |
> > | LIVE (ViT-B) | Specific Query | 67.4 | 69.5 |

---

> > > ### Author Response · Authors · 2025-12-03
> > > **More reply**
> > >
> > > **Q1:Clarifying Conceptual Novelty**
> > >
> > > We clarify that the novelty of LIVE is not a modulation pathway, but in the information flow pipeline that uses language information to help vision representation learning. While we utilize standard architectural blocks, our core contribution lies in shifting the paradigm from static vision alignment to adaptive vision computation. Unlike prior work that aligns an image to a single caption, LIVE utilizes dense, query-conditional pairs for each image. This data structure forces the vision backbone to dynamically compute features based on the specific query—effectively turning the encoder into a conditional computation engine rather than a static feature extractor. Thus, the method is not reducible to prior adapter-based work; we achieve this adaptive capability specifically through semantic representation matching across diverse queries, rather than standard global image-text alignment.
> > >
> > > **Q2: Data Contamination and Decontamination**
> > >
> > > Yes, by design. The training set (ImageNet) and evaluation sets (GQA, MMVP) are disjoint datasets. The strong performance on human-annotated GQA (+7.9 points) and MMVP (+34 points) confirms that our model generalizes to human-grounded data and does not rely on artifacts from the Gemini generation process.  See L310.
> > >
> > > **Q3: Use of Model-Generated “Oracle” Labels**
> > >
> > > Rationale: We do not treat Gemini outputs as absolute ground truth. Instead, they serve as a knowledge source. The metrics on “†” datasets are strictly intended to measure the knowledge transfer gap (i.e., how effectively LIVE absorbs the teacher's knowledge), rather than to serve as a proxy for human-labeled evaluation.
> > >
> > > Alignment with Human Judgement: Since there are no available human VQA annotations in the “†” benchmarks (ie, Imagenet, Caltech101, SUN, RefCOCO), we visually inspected Gemini-generated triplets and found they are all correct so far in the query-answer logic.  While we do not claim Gemini is a perfect human oracle, we qualitatively find it is a high-quality teacher for training. For evaluation against "ground truth," we point again to our strong results on the human-labeled MMVP and GQA datasets.
> > >
> > > **Q4: Scale and Fairness of Comparisons**
> > >
> > > 1. Source of Baselines: We confirm that all baseline results are taken directly from their original publications; we did not retrain them.
> > >
> > > 2. Compute and Data Volume: Our method, LIVE, achieves state-of-the-art performance on MMVP while using orders of magnitude less compute and data than leading baselines. The computation budget is roughly proportional to $\text{Data Size} \times \text{Model Size}$.
> > >
> > > vs. BRAVE (SOTA): We achieve +34.3% higher accuracy using only ~0.3% of the training data volume and <10% of the model parameters.
> > >
> > > vs. LLaVA: While we use more pre-training data, our model size is ~14x smaller, resulting in a significantly lower inference cost and total compute budget.
> > >
> > > | Method | Data Size | Model Size | MMVP Accuracy | Note |
> > > | :--- | :--- | :--- | :--- | :--- |
> > > | **Ours (LIVE)** | **16.4M** | **0.89B** | **76.3** | **High efficiency** |
> > > | MetaCLIP | 2.5B | 0.99B | 25.2 | |
> > > | BRAVE | Billions* | 10.3B | 42.0 | *LAION-2B, JFT-3B, WebLi |
> > > | LLaVA | 0.75M | 13B | 31.3 | High model cost |
> > > | InstructBLIP | Unknown^ | 14.2B | 16.7 | Combined 26 datasets |

---

### Meta-Review · Area_Chair_YPBg · 2026-01-07

**Summary:**

This paper received mixed reviews. The main concerns raised in the reviews are:
1. inadequate discussion of existing paradigms (`LBmL`).
2. no mitigation against data contamination when generating training data (`LBmL`).
3. unfair baseline comparisons due to training size differences (`LBmL`).
4. inadequate ablation studies (`LBmL`, `pP9q`).
5. performance gain stemming from data rather than model (`k7wj`).
6. potential spurious performance gain owing to text query shortcuts (`pP9q`).
7. the results presented only involve simple questions (`mP9X`).

Overall, all reviewers find the proposed method well-motivated, original, and effective. Many questions have been raised regarding the evaluation, related work discussion, and technical details. A strong rebuttal is provided presenting new results and compelling arguments. I believe most of the concerns have been addressed and the submission is now much stronger. Hence, I recommend Accept.

**Reviewer Concerns:**

The authors provided a very strong rebuttal with many new results and well-articulated, compelling arguments. I believe most of the concerns raised have been adequately addressed.

**Reviewer Scores:**

1. Reviewer `LBmL` (4->6+): it is likely that the reviewer might increase their rating as I believe most of their concerns have been adequately addressed in the rebuttal.
2. Reviewer `pP9q` (4->6+): it is likely that the reviewer might increase their rating as I believe most of their concerns have been adequately addressed in the rebuttal.
3. Reviewer `k7wj` (4->6?): the reviewer might increase their rating as I believe most of their concerns have been adequately addressed in the rebuttal.
4. Reviewer `mP9X` (8->8+): the reviewer might maintain or increase the rating as I believe their questions have been sufficiently answered in the rebuttal.

---

### Decision · Program_Chairs · 2026-01-26

Accept (Poster)